# Discovering Opinion Intervals
# from Conflicts in Signed Graphs

**Peter Blohm**∗
Aalto University
Espoo, Finland
peter.blohm@aalto.fi

**Florian Chen**∗
University of Oxford
Oxford, UK
florian.chen@cs.ox.ac.uk

**Aristides Gionis**
KTH Royal Institute of Technology
Digital Futures
Stockholm, Sweden
argioni@kth.se

**Stefan Neumann**
TU Wien
Vienna, Austria
stefan.neumann@tuwien.ac.at

## Abstract

Online social media provide a platform for people to discuss current events and exchange opinions with their peers. While interactions are predominantly positive, in recent years, there has been a lot of research to understand the conflicts in social networks and how they are based on different views and opinions. In this paper, we ask whether the conflicts in a network reveal a small and interpretable set of prevalent opinion ranges that explain the users' interactions. More precisely, we consider signed graphs, where the edge signs indicate positive and negative interactions of node pairs, and our goal is to infer opinion intervals that are consistent with the edge signs. We introduce an optimization problem that models this question, and we give strong hardness results and a polynomial-time approximation scheme by utilizing connections to interval graphs and the CORRELATION CLUSTERING problem. We further provide scalable heuristics and show that in experiments they yield more expressive solutions than CORRELATION CLUSTERING baselines. We also present a case study on a novel real-world dataset from the German parliament, showing that our algorithms can recover the political leaning of German parties based on co-voting behavior.

## 1 Introduction

Online social networks are essential parts of modern societies and are used by billions of people to discuss current events. Even though a majority of the interactions on such networks are positive, there are a substantial number of conflicts, particularly due to tensions among people with differing viewpoints [48, 47].

As a result, gaining a deeper understanding of these conflicts has become essential. This question is often studied using *signed graphs* [29, 48, 47], where each edge has a sign that is either positive (+) if two nodes interact amicably, or negative (−) if the interaction is conflicting. A classic formulation used to analyze signed graphs and gain insights about the graph structure and potentially the opinions of nodes is the CORRELATION CLUSTERING problem [4]. In CORRELATION CLUSTERING, we ask to partition the nodes of a given signed graph into clusters, so as to maximize the number of edges that are consistent with the clustering (or minimize the number of inconsistent edges).

---

∗ Equal contribution. This work was done while the authors were at TU Wien.

39th Conference on Neural Information Processing Systems (NeurIPS 2025).

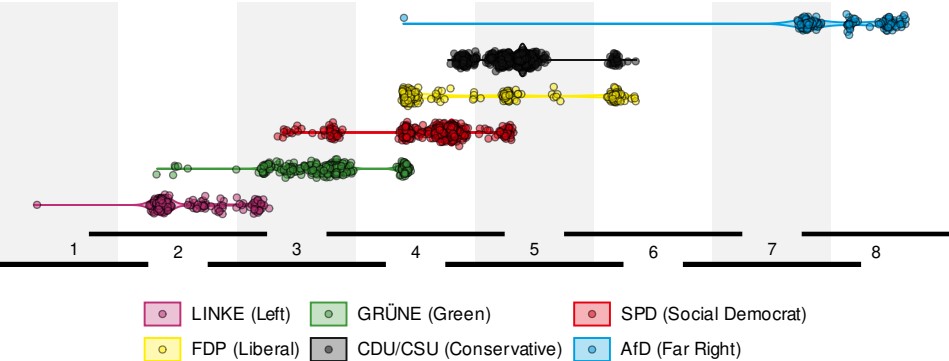

Figure 1: Visualization of our results on a signed graph based on co-voting behavior in the German parliament. We computed a solution to BEST INTERVAL APPROXIMATION with 8 intervals, where consecutive intervals overlap (the intervals are visualized at the bottom of the figure). Each point corresponds to a German politician and assignments are visualized by mapping them to their party (y-axis) and interval (x-axis); for example, interval 3 contains politicians from SPD, GRÜNE, and LINKE. In each interval, points are shifted left or right, based on the number of positive and negative edges the corresponding politician shares with adjacent intervals.

One drawback of the CORRELATION CLUSTERING formulation is that it makes hard decisions for the assignment of nodes into clusters and does not allow for a nuanced model in the presence of complex node interactions. For instance, in the landscape of European political parties, the opinions of representatives typically do not align perfectly with party lines; instead, members of ideologically-neighboring parties may agree on certain issues, while at the same time, members of the same party may disagree on other issues. Similar observations have been made for the US House of Representatives [2], however, when more parties are involved, modeling the interactions between representatives becomes increasingly complex.

In this paper, we introduce a novel problem to analyze signed graphs and discover structure that explains the nodes' interactions (conflicts and agreements) more accurately. Instead of assigning nodes to disjoint clusters, we seek to assign nodes to a small number of potentially-overlapping *opinion intervals*. The resulting structure can lead to meaningful insights and intuitive visualization (e.g., see Figure 1). We show that our problem is more expressive than CORRELATION CLUSTERING, thus resolving the drawback mentioned above. At the same time, our problem only requires the edge signs in the network as input, making it widely applicable.

**Our results.** First, we introduce the BEST INTERVAL APPROXIMATION problem: Given a signed graph $G = (V, E^+ \cup E^-)$, assign an interval $I_v \subset \mathbb{R}$ to every vertex $v \in V$ such that we maximize the number of edges $\{u, v\} \in E^+$ with $I_u \cap I_v \neq \emptyset$ and $\{u, v\} \in E^-$ with $I_u \cap I_v = \emptyset$. In other words, if two nodes are connected by a positive edge, then their corresponding intervals should overlap, whereas if they are connected by a negative edge, then their intervals should be disjoint. Note that for a node $v$, we can think of $I_v$ as the range of opinions that are acceptable to $v$ and yield an amicable interaction; all opinions outside of $I_v$ are not acceptable and yield a conflict. This problem is more expressive than the CORRELATION CLUSTERING problem of Bansal et al. [4], and a related problem by Kermarrec and Thraves [34] as we explain below.

Second, we show that BEST INTERVAL APPROXIMATION is NP-hard *even when the graph $G^+ = (V, E^+)$ induced by the positive edges forms a cycle*. In a sense, this is the strongest possible hardness result one could hope for since removing a single edge from the cycle $G^+ = (V, E^+)$ produces a path, for which intervals can always be assigned without any error. This implies that (unless P = NP) for BEST INTERVAL APPROXIMATION there is no FPT algorithm that parameterizes by the number of required edge removals and that the disagreement version of BEST INTERVAL APPROXIMATION cannot be approximated within any multiplicative factor. It also provides novel insights into the hardness of finding forbidden induced subgraphs, which rules out several algorithm design approaches. Our reduction is based on a result of Cygan et al. [22], but making it work for cycles requires several new ideas and gets significantly more complicated. We provide an overview of the reduction in Section 2.1.

Third, we consider a constrained version of BEST INTERVAL APPROXIMATION, where we are given a complete signed graph and a parameter $\varepsilon > 0$. Now, we are only allowed to use $k$ distinct intervals and each node must be assigned to one of them. This provides highly interpretable insights since the number of intervals is small. For this problem, we provide a polynomial-time approximation scheme (PTAS); specifically, we present an algorithm that computes a $(1 + \varepsilon)$-approximation in time $2^{O(k^2 \log(k/(\varepsilon\delta))/\varepsilon^3)} \cdot n$. This generalizes an algorithm by Giotis and Guruswami [28] that was developed for CORRELATION CLUSTERING with a fixed number of clusters. We provide an overview of the PTAS in Section 2.2.

Fourth, from a practical point of view, we introduce heuristics that we describe in Section 3. Our heuristics are inspired by the PTAS above and include several practical improvements. Our experiments find that BEST INTERVAL APPROXIMATION is substantially more expressive than CORRELATION CLUSTERING and that our heuristic algorithms succeed in exploiting this expressivity. On 8 real-world datasets, our methods find overlapping opinion interval assignments that represent the data with 38% fewer disagreements on average compared to CORRELATION CLUSTERING solutions found by state-of-the-art methods. This holds even when we use only 8 intervals, showing that already a small number of intervals yields expressive and interpretable representations.

Furthermore, we perform a case study on a novel dataset based on co-voting behavior in the German parliament, which we make publicly available. The output of our algorithm allows us to reconstruct the leaning of the political parties, as we demonstrate in Figure 1. Besides accurately reflecting the German political spectrum, the figure also reveals the coalition governments throughout the past decade (see Section 4). We stress that, due to the overlapping spectrum from the left to the right, finding such a structure would not be possible in existing problems like CORRELATION CLUSTERING.

We conclude the paper with several interesting questions for further research in Section 5, and present our proofs and additional experimental results in the supplementary material.

**Related work.** The BEST INTERVAL APPROXIMATION problem is closely related to the SITTING ARRANGEMENT problem by Kermarrec and Thraves [34]: Given a signed graph $G = (V, E^+ \cup E^-)$, can we assign a vector $x_u \in \mathbb{R}^\ell$ to each $u \in V$ such that for all positive edges $\{u, v\} \in E^+$ and negative edges $\{u, w\} \in E^-$ the inequality $\|x_u - x_v\|_2 < \|x_u - x_w\|_2$ holds? Kermarrec and Thraves [34] presented several results for the case of $\ell = 1$, i.e., embedding $G$ into the real line. Cygan et al. [22] improved upon this and showed that for a complete signed graph $G$ such an assignment exists if and only if the subgraph induced by its positive edges $G^+ = (V, E^+)$ is a unit interval graph [59]. Given this characterization, we note that our problem is more expressive since we allow general (non-unit) intervals. Besides these theoretical insights, Pardo et al. [55, 56] provided heuristics for an optimization version that aims to minimize the number of violated constraints on the vectors $x_u$ above. However, this objective is substantially different from ours and thus incomparable.

As mentioned before, CORRELATION CLUSTERING [4] is highly related to our work and is stated as follows: Given a signed graph $G = (V, E^+ \cup E^-)$, partition its vertices into disjoint clusters $C_1, \ldots, C_k \subseteq V$ such that the number of positive edges within the clusters $C_i$ and the number of negative edges between different clusters $C_i$ and $C_j$, $i \neq j$, is maximized. Here, the value of $k$ can be picked by the algorithm. CORRELATION CLUSTERING has received a lot of attention in the past two decades in social network analysis and image segmentation, spanning approximation algorithms [13, 60, 28, 20, 18, 17], more expressive formulations [7], and results in dynamic, online, parallel, and streaming settings [16, 15, 19, 3, 41, 51, 43]. There has also been continued interest in developing heuristics [50, 1, 61, 66, 42, 6, 9]. We refer to the book by Bonchi et al. [8] for more references.

Another closely related problem is that of (Unit) Interval Editing. Specifically, in (Unit) Interval Editing, the task is to transform an unsigned graph into a (unit) interval graph using a minimum number of edge deletions and insertions. This problem is known to be NP-hard already since the seminal work of Garey and Johnson [27] and it is fixed-parameter tractable (FPT) when parameterized by the number of edge insertions and deletions [11, 33, 64, 12]. We further discuss the relation of BEST INTERVAL APPROXIMATION to these problems in Section 2.

Opinion formation models, such as the DeGroot model [24], the Friedkin–Johnsen model [26], or the bounded-confidence model [38, 23], are also related. These models have recently received a significant amount of attention in computer science and machine learning [53, 65, 67, 62] and assign a real-valued opinion to each node in a graph, which allows a more fine-grained understanding of conflicts than CORRELATION CLUSTERING. However, estimating the parameters of such models is highly

challenging and requires more information than the edge signs of a signed graph [5, 45, 44]. Thus, our method is more easily applicable as it requires substantially less (and particularly less sensitive) data.

In relation to our case study on the German parliament, the (DW)-NOMINATE algorithm [57, 58, 52] also predicts ideological positions of legislators based on co-voting data. It models legislators and roll-call votes as a signed bipartite graph and applies maximum-likelihood estimation to infer each legislator's ideological location in a low-dimensional Euclidean space, together with a Gaussian utility function centered at that point. However, since (DW)-NOMINATE operates on a bipartite graph, it is not applicable in more general social network settings where the input is given as a unipartite graph. This is in contrast to our methods, which only require a unipartite signed graph as input.

Several related works consider versions of the CORRELATION CLUSTERING objective for partitioning signed graphs to reveal community structures [21, 40, 14, 54, 63]. However, similar to CORRELATION CLUSTERING, these methods do not allow finding overlapping communities. Thus, they cannot explicitly consider individual tolerance of other opinions in a way comparable to opinion intervals.

Further, Dwork et al. [25] employed opinion intervals to analyze content moderation in online communities. They use them to study the effectiveness of moderation strategies on online platforms.

**Preliminaries.** A *signed graph* $G = (V, E^+ \cup E^-)$ is given by its vertices $V$, positive edges $E^+$, and negative edges $E^-$, where $E^+ \cap E^- = \emptyset$. It is *complete* if $E^+ \cup E^- = \binom{V}{2}$. For $u \in V$, we write $N^+(u)$ to denote its neighbors in $E^+$ and $N^-(u)$ to denote its neighbors in $E^-$.

A graph $G = (V, E)$ is an *interval graph* if we can assign an interval $I_v \subset \mathbb{R}$ to all vertices $v \in V$ such that for all $u, v \in V$, it holds that $\{u, v\} \in E$ if and only if $I_u \cap I_v \neq \emptyset$. Additionally, we say that $G$ is a *unit* interval graph if all intervals have length 1.

## 2 Problem Definition and Theoretical Results

In this section, we define our novel problem and state our main theoretical results.

**Problem 2.1** (BEST INTERVAL APPROXIMATION)**.** Given a signed graph $G = (V, E^+ \cup E^-)$, find a set $\mathcal{I} = \{I_v \subset \mathbb{R} \colon v \in V\}$ of non-empty, contiguous intervals that maximizes

$$\mathsf{agree}(G, \mathcal{I}) = \sum_{\{u,v\} \in E^+} \mathbb{1}(I_u \cap I_v \neq \emptyset) + \sum_{\{u,v\} \in E^-} \mathbb{1}(I_u \cap I_v = \emptyset), \tag{1}$$

where $\mathbb{1}(E)$ is indicator function, which takes value 1 if $E$ is true and 0 otherwise.

Intuitively, the problem assigns an interval $I_v$ to each vertex $v$ and asks that two intervals overlap if their corresponding vertices are connected with a positive edge and do not overlap if their vertices are connected with a negative edge. To connect this problem to opinions, we may consider an interval $I_v$ for a node $v$ as the range of opinions that are acceptable to $v$. The length $|I_v|$ can further be seen as a measure of $v$'s tolerance towards the opinion spectrum to the left and to the right.

We will refer to the formulation in Problem 2.1 as the *agreement* version of BEST INTERVAL APPROXIMATION, which asks to satisfy as many edges as possible. We will also talk about the *disagreement* version, which aims to minimize the number of edges violating the interval assignment. Their complexity is the same for exact solutions, but they differ w.r.t. approximation guarantees.

*Relationship to* CORRELATION CLUSTERING. Next, we observe that BEST INTERVAL APPROXIMATION is more expressive than CORRELATION CLUSTERING: First, consider any CORRELATION CLUSTERING solution $C_1, \ldots, C_k$, and consider $k$ non-overlapping intervals $I_1, \ldots, I_k$. Now we assign each vertex in $C_i$ to the same interval $I_i$. Thus, if $u, v \in C_i$, then their intervals overlap, and, if $u \in C_i$ and $v \in C_j$ for $i \neq j$, then their intervals do not overlap. This implies that the optimal solution of BEST INTERVAL APPROXIMATION will always yield an agreement at least as large as for CORRELATION CLUSTERING. Second, our interval representation is strictly more expressive and, for instance, allows us to model non-transitive node relationships and this property is illustrated in Figure 2. This is neither possible in CORRELATION CLUSTERING nor in the structural balance

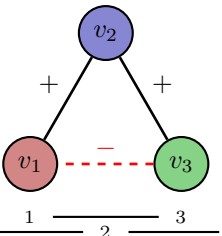

Figure 2: A triangle with one negative edge and its (exact) interval representation.

theory of Harary [29], in both of which no cycle with exactly one
negative edge can be represented without error.

*The case of complete graphs and relationship to Interval Editing.* If $G$ is a complete signed graph then
it can be represented without error in BEST INTERVAL APPROXIMATION if and only if $G^+ = (V, E^+)$
is an interval graph. That is because missing edges in $G^+$ correspond to negative edges in $G$ (since
$G$ is complete). Thus, making the minimum number of edge deletions/insertions to turn $G^+$ into
an interval graph is equivalent to flipping the minimum number of edge signs in $G$ such that we
have agreement for all edges. Hence, for complete graphs, we can rely on the rich literature on
Interval Editing which asks for the minimum number of edge changes to $G^+$ such that it becomes an
interval graph. The results of Cao [10] now imply that BEST INTERVAL APPROXIMATION is FPT for
complete graphs when only allowing a fixed number of sign changes (in one direction). However, in
social networks this number will be large for real-world instances and thus these algorithms are not
applicable in practice. Furthermore, our hardness results show that such FPT results are not possible
in incomplete graphs when parameterized by the number of required edge deletions (see Section 2.1).

## 2.1 Computational hardness

Next, we show that BEST INTERVAL APPROXIMATION is NP-hard. We show this by using a
reduction from the NP-complete problem ACYCLIC DIGRAPH PARTITION [22], where we are given
a directed graph $H = (V, E)$ and have to decide whether one can partition $V$ into two sets $V_1$ and $V_2$,
such that both $H[V_1]$ and $H[V_2]$ are directed acyclic graphs.

Our hardness result is stated below. In the theorem, we say that an interval representation is *conflict-free* if it achieves agreement for all edges, i.e., if Equation (1) equals the number of edges in the
graph. Further, we will consider the minimum number of *edge deletions* required to make the graph
conflict-free, which is identical to the optimal objective function value for the disagreement version
of BEST INTERVAL APPROXIMATION.[1]

**Theorem 2.2.** *There exists a polynomial-time algorithm that, given an instance $H = (V, E)$ of*
ACYCLIC DIGRAPH PARTITION, *outputs an instance $G = (V', E^+ \cup E^-)$ of* BEST INTERVAL
APPROXIMATION *with the following properties: (1) $H$ is a YES-instance if and only if a conflict-free
interval representation of $G$ exists. (2) If $H$ is a NO-instance, then only a single edge deletion is
required to obtain a conflict-free interval representation of $G$. (3) $|V'| = \mathcal{O}(|V|)$, $|E^+ \cup E^-| < \binom{|V'|}{2}$,
and $G^+ = (V', E^+)$ is a cycle. Thus,* BEST INTERVAL APPROXIMATION *is* NP-hard.

The theorem has several implications for BEST INTERVAL APPROXIMATION in incomplete graphs:
(1) The disagreement version is hard to approximate within any factor. (2) It is not FPT when
parameterized by the number of required edge deletions (unless $P = NP$), separating it from the
problem in complete graphs. (3) The result holds even when restricted to graphs $G = (V', E^+ \cup E^-)$
where $G^+ = (V', E^+)$ is a chordless cycle. This is intriguing because many algorithmic results on
interval graphs rely on detecting forbidden induced subgraphs like chordless cycles of four or more
vertices [10, 64, 37]. Our hardness result implies that detecting these forbidden structures is NP-hard
for incomplete signed graphs.

We prove Theorem 2.2 in Appendix A, where we construct a new graph $G$ from an ACYCLIC
DIGRAPH PARTITION instance $H$, and show that $G$ can be represented conflict-free if and only if $H$
is a YES-instance. In $G$, we introduce two auxiliary vertices $L$ and $R$ and we show that all vertices
whose intervals overlap with the interval of $L$ ($R$) must be in partition $V_1$ ($V_2$) in the optimal solution
of ACYCLIC DIGRAPH PARTITION. Thus, the overlap structure of the intervals encodes a partition
of the vertices of $H$. Crucially, we use negative edges to enforce a topological ordering over these
partitions and the induced subgraphs $H[V_1]$ and $H[V_2]$, and we introduce further auxiliary vertices
to ensure that $G$ forms a cycle.

## 2.2 A PTAS for fixed $k$ in complete graphs

From an algorithmic perspective, we provide a PTAS when $G$ is a complete graph and when each
vertex must be assigned to one of $k$ intervals, where $k = O(1)$. Formally, we study a version of BEST

---

[1]This is the case since minimizing the number of edges violating the interval assignment is equivalent to
deleting a minimum number of edges such that for the remaining graph (after the edges were deleted) there
exists a conflict-free interval representation.

INTERVAL APPROXIMATION in which we must find $k$ intervals $I_1, \ldots, I_k \subset \mathbb{R}$ and each vertex $v \in V$ must be assigned to one of these intervals. In practice, the small number of intervals makes the results highly interpretable. Additionally, it applies to scenarios such as analyzing political votes, where we would like to have one interval representing each party, and the number of parties is small. Our result for this restricted version of the problem is as follows.

**Theorem 2.3.** *Let $G$ be a complete signed graph and let $\varepsilon > 0$, $\delta > 0$ and $k \in \mathbb{N}$ be parameters. There exists an algorithm that, with probability at least $1 - \delta$, returns a $(1 + \varepsilon)$-approximate solution for* BEST INTERVAL APPROXIMATION *when the algorithm can only use $k$ different intervals and it runs in time $2^{O(k^2 \log(k/(\varepsilon\delta))/\varepsilon^3)} \cdot n$.*

The complete description and analysis of the PTAS are provided in Appendix B. An overview to obtain this result is as follows. Since $k$ is fixed, we can enumerate all possible choices of $k$ intervals with respect to their overlap structure. Now, given a fixed set of $k$ intervals, the main observation is that this corresponds to a generalized instance of CORRELATION CLUSTERING where we are given $k$ fixed clusters that might overlap. Specifically, when two clusters $V_i$ and $V_j$ overlap, we want their vertices to be connected by positive edges (rather than negative edges in the classic version of CORRELATION CLUSTERING). Then, we show that we can generalize a PTAS from Giotis and Guruswami [28] as described below.

We solve the generalized version of CORRELATION CLUSTERING by partitioning $V$ into $m = O(1/\varepsilon)$ equally-sized subsets $V_1, \ldots, V_m$. Then, for each $i = 1, \ldots, m$, we proceed as follows. We sample a set of vertices $S_i \subseteq V \setminus V_i$ of size $\tilde{O}\left(1/\varepsilon^2\right)$. Now, we enumerate all possible assignments of $S_i$ into $(S_{i,1}, \ldots, S_{i,k})$, where $S_{i,\ell} \subseteq S_i$ are the vertices assigned to interval $I_\ell$, and for each such assignment, we greedily assign the vertices $v \in V_i$ to the interval that maximizes the agreement of $v$'s edges to the clustering of $S_i$ given by $(S_{i,1}, \ldots, S_{i,k})$. This process gives a clustering of $V_i$ and we show how the clusterings of the $V_i$ can be merged to obtain a global clustering of $V$.

Our analysis is similar to that of [28] and shows that the sets $S_i$ are small enough such that enumerating all assignments is not too expensive, and simultaneously large enough that for most vertices they give us a good estimate for the agreement of their edges w.r.t. a fixed clustering. This is the key to arguing that the greedy assignment will yield a good result when we consider the correct clustering of the $S_i$. In contrast to [28], we have to take into account the overlap of intervals when computing the estimates. As for [28], the approach does not extend to incomplete graphs or large $k$, since then the sets $S_i$ become too large and enumeration would not be possible anymore.

## 3   Heuristic Algorithms

Next, we present our heuristic *Greedy Agreement Interval Assignment* (GAIA) for BEST INTERVAL APPROXIMATION, which is given intervals $I_1, \ldots, I_k$ as input to which all vertices must be assigned.

We use the following notation. For an interval $I_\ell$ we let $\mathsf{overlap}(\ell) = \{\ell' : I_\ell \cap I_{\ell'} \neq \emptyset\}$ denote the set of intervals $I_{\ell'}$ that overlap with $I_\ell$. Furthermore, we will consider disjoint vertex clusters $C_1, \ldots, C_k \subseteq V$ that correspond to an assignment of the vertices to the intervals, i.e., $C_\ell$ contains all vertices assigned to interval $I_\ell$. Now, for a vertex $u$ and $C_1, \ldots, C_k$ as before, we write

$$\mathsf{agree}(u, \ell, (C_1, \ldots, C_k)) = \sum_{\ell' \in \mathsf{overlap}(\ell)} \left| N^+(u) \cap C_{\ell'} \right| + \sum_{\ell' \notin \mathsf{overlap}(\ell)} \left| N^-(u) \cap C_{\ell'} \right|$$

for the number of agreeing edges when assigning vertex $u$ to interval $I_\ell$ for the clustering $C_1, \ldots, C_k$.

Now, we describe GAIA and state its pseudocode in Algorithm 1. GAIA is based on *iterative refinement*: After computing an initial greedy assignment of all vertices, the solution is improved by reassigning vertices in multiple epochs. This reassignment procedure is carried out in batches to avoid local minima. In each epoch, the vertex set is partitioned into random batches $V_1, \ldots, V_m$, and the algorithm iterates over these batches one at a time. When processing a batch $V_i$, all vertices in the batch are first unassigned and then reassigned using the greedy procedure described below. This can be viewed as a practical version of PTAS from Theorem 2.3, where, instead of brute-forcing solutions on out-of-batch vertices, the algorithm leverages the previously constructed greedy solution.

The core of GAIA is the *greedy assignment* of vertices in $V_i$ to intervals in Line 5–8 in Algorithm 1. Here, each vertex $v \in V_i$ is assigned to the interval $I_\ell$ (and its corresponding cluster $C_\ell$) that

**Algorithm 1:** Greedy Agreement Interval Assignment (`GAIA`)

---

**Input:** Signed graph $G = (V, E^+ \cup E^-)$, intervals $I_1, \ldots, I_k$
**Output:** Interval assignment $(C_1, \ldots, C_k)$ where $C_\ell$ are the vertices assigned to interval $I_\ell$

1 Compute an initial assignment of the vertices to the intervals;
2 **for each** epoch **do**
3      Randomly partition $V$ into $m$ sets $V_1, \ldots, V_m$ of size $\frac{n}{m}$ each;
4      **for** $i = 1, \ldots, m$ **do**
5          $C_\ell \leftarrow C_\ell \setminus V_i$ for all $\ell = 1, \ldots, k$;          `// Unassign all vertices in V_i`
6          **for** $v \in V_i$ *in order of maximum agreement* **do**
7              $\ell \leftarrow \operatorname{argmax}_{\ell=1\ldots k} \mathsf{agree}(v, \ell, (C_1, \ldots, C_k))$;
8              $C_\ell \leftarrow C_\ell \cup \{v\}$;          `// Assign v to I_ℓ`
9 **return** $(C_1, \ldots, C_k)$;

---

maximizes $\mathsf{agree}(v, \ell, (C_1, \ldots, C_k))$ (breaking ties at random). Crucially, we assign the vertices with the highest agreement values first, as these vertices are easier to assign and their assignment provides more information when assigning later vertices.

We also provide a version of `GAIA` called *Variable ENergy Uphill Search* (`VENUS`), which additionally uses *simulated annealing* [36] to further increase the variability of its solutions. In `VENUS`, vertices are not necessarily assigned to the interval that maximizes $\mathsf{agree}(v, \ell, (C_1, \ldots, C_k))$, but instead, each vertex is assigned to an interval selected probabilistically according to a temperature-scaled softmax distribution over agreement values. To that end, Line 7 of Algorithm 1 is replaced with $\ell \sim \operatorname{softmax}_{\ell=1\ldots k} \frac{\mathsf{agree}(v,\ell,(C_1,\ldots,C_k))}{t}$. Here, $t$ is a temperature parameter and controls the level of randomness during the assignment. A temperature $t$ close to 0 corresponds to a more greedy approach, while higher temperatures lead to increasingly uniform random assignments. The annealing schedule follows exponential decay: the temperature is initialized at $t_0$ and multiplied by a decay factor $\alpha \in (0, 1)$ after every $\tau$ epochs. This gradually reduces randomness and encourages convergence.

## 4 Experiments

Next, we experimentally evaluate our algorithms. Our code is available in a GitHub repository.[2] We aim to answer the following research questions:

(RQ1) Does BEST INTERVAL APPROXIMATION yield a substantial increase in expressiveness compared to CORRELATION CLUSTERING?

(RQ2) How computationally efficient and scalable are our proposed algorithms?

(RQ3) What is the trade-off between solution quality and the number of intervals?

(RQ4) Are the solutions produced by our method interpretable?

(RQ5) Are our algorithms able to recover ground-truth interval structures?

We evaluate our algorithms on real-world datasets from SNAP [46] and KONECT [39]. We further provide a novel dataset based on voting data from the German Bundestag (parliament) between the years of 2012 and 2025 and make it available in our repository. In this dataset, each Bundestag member corresponds to a vertex in the graph, and two members are connected by a positive edge if they vote the same way in at least 75% of the sessions they both attended. Conversely, they are connected by a negative edge if their votes align in 25% of sessions or less.

In our experiments, we evaluate our base algorithm, `GAIA`, as well as the `VENUS` variant that uses simulated annealing. For `VENUS`, we use an initial temperature of 100 and a decay factor of $2/3$, applied every 5 epochs. Both are run with 10 batches for vertex reassignment. For the interval structure that our algorithms receive as input, unless stated otherwise, we use a chain-like structure of 8 intervals, where each interval overlaps with the next, e.g., $[0, 1], [1, 2], \ldots, [7, 8]$, and we call this interval structure an `8-Chain`. This structure was chosen to find a trade-off between increased

---

[2]https://github.com/Peter-Blohm/discovering_opinion_intervals

Table 1: Overview of the best solutions found by the algorithms. Reported is the percentage (%) of edges violated (lower is better). Our algorithms use the `8-Chain` interval structure while the CORRELATION CLUSTERING algorithms use an unrestricted number of clusters.

| Dataset | $|V|$ | $|E|$ | $\frac{|E^+|}{|E|}$ | Our algorithms | | CORRELATION CLUSTERING baselines | | | | Improvement |
|---|---|---|---|---|---|---|---|---|---|---|
| | | | | GAIA | VENUS | GAEC | GAECKLj | SCMLEvo | RAMA | |
| BitcoinOTC | 5 881 | 21 434 | 0.85 | **3.32** | 3.55 | 5.58 | 5.57 | 5.57 | 5.64 | 40.39% |
| Chess | 7 301 | 32 650 | 0.58 | 19.82 | **19.63** | 28.64 | 28.10 | 27.33 | 39.98 | 28.17% |
| WikiElec | 7 115 | 100 355 | 0.78 | **11.24** | 11.26 | 14.13 | 14.13 | 14.13 | 14.45 | 20.45% |
| Bundestag | 1 480 | 397 497 | 0.81 | **0.25** | **0.25** | 3.06 | 2.95 | 2.95 | 3.72 | 91.53% |
| Slashdot | 82 140 | 498 532 | 0.76 | 9.05 | **8.94** | 13.75 | 13.66 | 13.52 | 17.17 | 33.88% |
| Epinions | 131 580 | 708 507 | 0.83 | 4.47 | **4.42** | 6.83 | 6.68 | 6.67 | 6.86 | 33.73% |
| WikiSigned | 138 587 | 712 337 | 0.88 | 4.94 | **4.85** | 6.17 | 6.17 | 6.17 | 6.96 | 21.39% |
| WikiConflict | 116 836 | 2 014 053 | 0.38 | 3.44 | **3.43** | 5.87 | 5.82 | 5.82 | 6.02 | 41.06% |

expressivity and intuitive interpretation (see also discussion of RQ3 below). Where applicable, experiments were repeated 50 times on different random seeds, and standard deviations are reported.

Numerous approaches have been proposed for solving CORRELATION CLUSTERING in social networks analysis [49, 9, 31, 30] and in computer vision [35, 66, 42, 1]. To provide a representative performance overview, we selected four state-of-the-art algorithms for comparison:

- `GAEC` [35]: A method that incrementally merges clusters to minimize disagreement.

- `GAECKLj` [35]: An extension of `GAEC` that additionally applies local search postprocessing.

- `SCMLEvo` [30]: An algorithm combining multilevel local search with evolutionary techniques.

- `RAMA` [1]: An algorithm using polyhedral relaxation and message passing to guide cluster merging.

For each of these algorithms, we run the authors' publicly available implementations. In contrast to our algorithms, which only use 8 intervals, the baselines may use an unrestricted number of clusters.

Throughout our experiments, we report the *disagreement*, i.e., the fraction of violated edges in solutions found across all real-world datasets (rather than the number of agreeing edges as in Equation (1)), as this makes the algorithms' performance easier to compare. Further details on the experiment setup, as well as additional results, are described in Appendix C.

*Expressivity analysis (RQ1).* To compare the expressivity of BEST INTERVAL APPROXIMATION with CORRELATION CLUSTERING, we run our algorithms and the baselines on the real-world datasets. The results in Table 1 show that our algorithms consistently find interval assignments that achieve 20% to 90% fewer disagreements than the best CORRELATION CLUSTERING solution. Across all datasets, our results have 38% less disagreement on average, even though our methods only use 8 intervals, whereas the CORRELATION CLUSTERING baselines use an unrestricted number of clusters. Hence, our heuristic algorithms manage to effectively use the increased expressivity of the overlapping interval structure. Additionally, we find that `VENUS` tends to outperform `GAIA` slightly, particularly for larger graphs.

*Computational efficiency (RQ2).* Next, we assess the runtime efficiency of our algorithms by tracking the progression of the objective value over time. Representative results for Slashdot are presented in Figure 3a. `GAIA` makes the most progress in the first 15 seconds, followed by slower, incremental gains. `VENUS` exhibits a similar pattern, though slightly delayed, likely due to its initially high temperature which slows early convergence. However, this high initial temperature appeared to be necessary to achieve improvement over `GAIA`'s results. In most instances, both heuristics stopped after 50 epochs without improvement in the first five minutes of runtime, with `GAIA` often terminating after a few seconds. The algorithms' running time until convergence scales approximately linearly in the number of edges and on all datasets our methods terminate within 30 minutes; we also elaborate on this in the appendix.

*Number of intervals (RQ3).* To investigate the relationship between the number of intervals and the solution quality, we ran our algorithms with 4, 8, 12, and 16 intervals. As for the `8-Chain`, in each interval structure, every interval overlaps with its successor and predecessor. In Figure 3a, we illustrate the convergence behavior, and Figure 3b presents the solution quality after convergence, both on the Slashdot dataset. Our results show that using only 4 intervals leads to poor solution quality

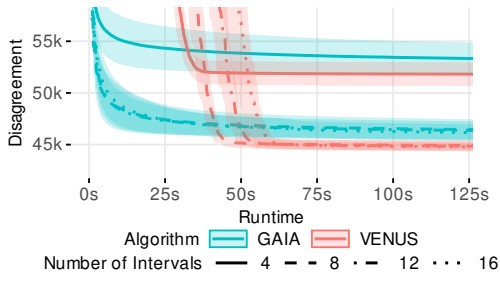
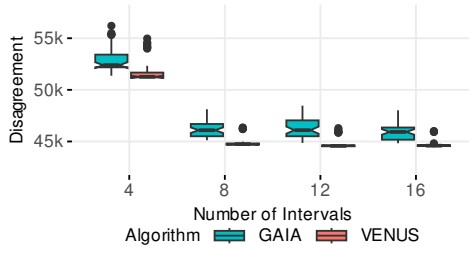

(a) Objective value over time.

(b) Final objective for different interval structures.

Figure 3: Performance of our algorithms with different configurations on the Slashdot dataset.

compared to the higher numbers, suggesting that such a limited structure may not adequately capture the complexity of the graph. While the solution quality improves with more intervals, 8 intervals seem sufficiently expressive for this graph, with only marginal improvements beyond that. This behavior is typical for other problem instances as well. Again, we see that VENUS tends to perform slightly better, and most notably, its results have much less variance compared to GAIA.

*Interpretability (RQ4).* To study the interpretability of our solutions, we perform a case study on the Bundestag dataset. We present a representative solution found by VENUS in Figure 1. As the dataset models co-voting behavior of politicians, we expect our interval representation to resemble the German political spectrum, and, indeed, this is the case. Our result assigns most members from the same party to the same or neighboring intervals. For each party, except the FDP, we can identify one interval consisting mainly of members of that party. We note that the slight splitting up of parties is natural due to government coalitions they formed throughout the years. Also, the behavior of the FDP can be traced back to different coalition governments they were part of (they formed governments with the conservative CDU/CSU, as well as with the left/center GRÜNE and SPD). We consider the ability of our algorithms to extract such highly overlapping structure as a substantial improvement over CORRELATION CLUSTERING, and this is also emphasized by the objective function values reported in Table 1, where our methods have 91% fewer disagreeing edges on this dataset.

*Reconstruction of ground-truth data (RQ5).* Next, we evaluate our algorithms on synthetic data. We fix the 8-Chain and generate a graph with $n = 800$ vertices as follows. We assign $\frac{n}{8}$ vertices to each interval, and we introduce edges with signs corresponding to the interval structure for $d\binom{n}{2}$ random pairs of vertices, where $d \in [0, 1]$ is the desired density of the graph. Each edge obtains a correct edge sign based on the interval structure with probability $1 - p$ and we flip the sign with probability $p$. In our experiments, we measure the relative change of the objective function achieved by our algorithms compared to the ground-truth assignment in percent, $\frac{\text{agree}(G, \text{ground truth}) - \text{agree}(G, \text{ALG})}{|E|} \cdot 100$, and we also report the accuracy with which vertices are assigned to their corresponding interval.

Figure 4 shows the result of our experiments. Without sign noise, the solutions are always within 6.5% of the ground truth for VENUS, and within 12.5% of the ground truth for GAIA. We also obtain a high accuracy in reconstructing the ground-truth assignment. With increasing sign noise, the true solution becomes increasingly suboptimal to the point where both GAIA and VENUS find alternative solutions, with *better* objective values than the ground truth (this is the case when we have negative y-axis values in the plot). This increased objective value, however, comes at the cost of less accuracy in the vertex assignment. The point at which alternative solutions become viable depends heavily on the density of the graph, with denser graphs being more resilient to this phenomenon.

## 5 Conclusion

We introduced the BEST INTERVAL APPROXIMATION problem and showed that it is more expressive than CORRELATION CLUSTERING, both theoretically and in experiments. We gave strong hardness results for incomplete graphs, as well as a PTAS for complete graphs and fixed $k$. We also provided efficient heuristics, which find interval assignments with significantly better objective values than CORRELATION CLUSTERING solutions found by state-of-the-art algorithms, and we showed that these interval assignments are highly interpretable.

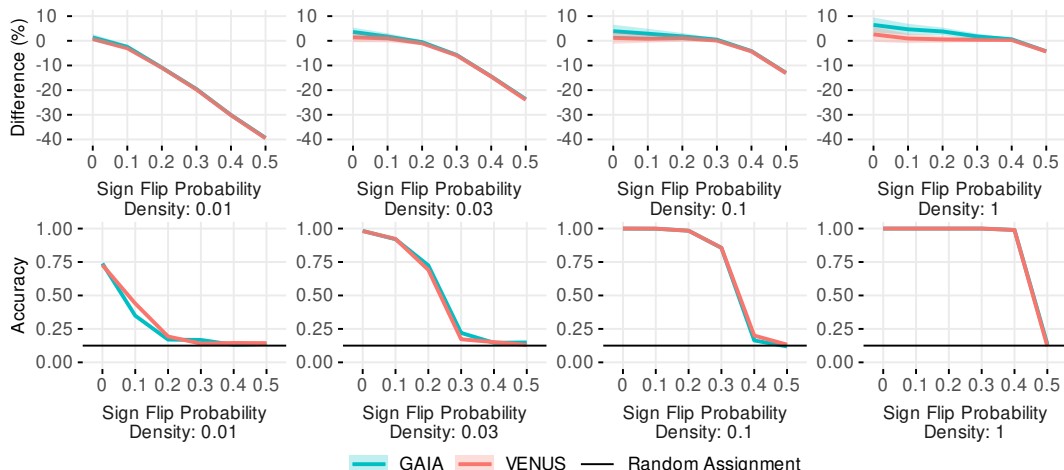

Figure 4: Results on synthetic data. For different edge density levels $|E| / \binom{|V|}{2}$, we report the normalized difference $\frac{\text{agree}(G,\text{ground truth}) - \text{agree}(G,\text{ALG})}{|E|}$ averaged over 50 runs, and standard deviations. Negative numbers indicate an improvement over the ground truth. Further, we report the accuracies of *the solution with the lowest disagreement*. GAIA and VENUS reconstruct the ground truth under considerable levels of noise in dense graphs, and find alternative, better solutions in sparse graphs.

We believe there are several interesting directions for future work, which we describe next.

From a more theoretical point of view, several problems remain unresolved. First, we conjecture that in the agreement version of the problem (for $k$ not fixed), the optimal solution can always satisfy a $\frac{3}{4}$-fraction of the edges. This claim is supported by ILP-solutions that we computed on small instances, and it is tight, for instance, when taking two cliques with negative edges and connecting each pair of their vertices with a positive edge. Second, it is interesting to study whether a PTAS exists in this setting. Third, our hardness results do not allow us to rule out that for fixed $k$ and complete graphs a PTAS exists for the disagreement version of our problem. While the techniques of Giotis and Guruswami [28] for CORRELATION CLUSTERING do not seem to extend to this setting, obtaining such a result would be interesting.

From a modeling perspective, several extensions are well-motivated. First, a natural extension is to move beyond one-dimensional intervals. Interestingly, neither our PTAS nor our heuristic algorithms are inherently restricted to intervals on a line. Rather, they only require knowledge of which clusters overlap and which do not. Hence, an empirical study using higher-dimensional intervals could allow a more nuanced discovery of opinions along multiple axes. Second, it might be interesting to consider temporal or dynamic settings in which opinion ranges expand or contract over time. Here, one could consider making the opinion intervals expand or contract depending on the nodes' centrality or the homophily of their immediate neighborhood.

From a machine learning perspective, it is interesting to study whether methods like GNNs can outperform our algorithms. This might be particularly promising when additional information, such as node labels, are available, which can be exploited by the GNNs.

## Acknowledgments and Disclosure of Funding

The authors thank the anonymous reviewers for their helpful comments, which have helped us to improve the presentation of the paper. We further thank Sebastian Lüderssen for useful discussions.

This research has been funded by the Vienna Science and Technology Fund (WWTF) [Grant ID: 10.47379/VRG23013], the ERC Advanced Grant REBOUND [834862], the Swedish Research Council (VR) [2024-05603], the European Commission MSCA DN ARMADA [101168951], and the Wallenberg AI, Autonomous Systems and Software Program (WASP) funded by the Knut and Alice Wallenberg Foundation.

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

# A  Hardness Result

In this section, we prove our hardness result from Theorem 2.2. In the proof, we use the notation $I = [\ell(I), r(I)]$, where $\ell(I)$ denotes the infimum and $r(I)$ denotes the supremum of the interval.

## A.1  Construction

We describe a reduction from an instance $H = (V, E)$ of ACYCLIC DIGRAPH PARTITION to an instance $G = (V', E^+ \cup E^-)$ of BEST INTERVAL APPROXIMATION, where $(V', E^+)$ forms a cycle.

First, the set of vertices $V'$ consists of:

1. Seven constant vertices $V'_c = \{S, L, M, R, T, H_S, H_T\}$. These vertices will be used to constrain the structure of the solution. With sets of negative edges, we will force any conflict-free representation to assign each vertex in $V$ to a sub-interval of either $L$ or $R$. The names of the vertices stand for start, left, middle, right, target, help-start, and help-target, respectively.

2. Four vertices for each vertex $v \in V$: $V'_v = \{M_v, A_v, X_v, B_v\}$, where $X_v$ corresponds to the original vertex $v$ in $H$ and the other vertices are used to structure the instance.

Next, we define the set of positive edges $E^+$ to form a cycle over $V'$. For this, we use an arbitrary ordering of $V = \{v_1, \ldots, v_n\}$, and construct $E^+$ as the union of the following sets:

1. $E^+_c = \{\{S, L\}, \{L, M\}, \{M, R\}, \{R, T\}, \{S, H_S\}, \{H_T, T\}\}$
2. for all $v \in V$: $E^+_v = \{\{M_v, A_v\}, \{A_v, X_v\}, \{X_v, B_v\}\}$
3. $E^+_V = \{\{H_S, M_{v_1}\}\} \cup \bigcup_{i \in [1, n-1]} \{\{B_{v_i}, M_{v_{i+1}}\}\} \cup \{\{B_{v_n}, H_T\}\}$

Finally, we construct $E^-$ as the union of the following sets:

1. $E^-_S = \{\{S, v'\} : v' \in V' \setminus \{L, H_S\}\}$, $E^-_T = \{\{T, v'\} : v' \in V' \setminus \{R, H_T\}\}$
   We connect negatively $S$ and $T$ to each vertex in the graph besides their positive neighbors. This forces $I_S$ and $I_T$ to be the outermost intervals in any conflict-free interval representation, as otherwise the interval of some negatively connected vertex intersects either of them. See Lemma A.1. To break symmetry, we assume without loss of generality that $r(I_S) < \ell(I_T)$.

2. $E^-_c = \{\{L, R\}\}$
   This edge ensures that $I_L$ and $I_R$ are disjoint, and in any conflict-free representation, $r(I_L) < \ell(I_R)$, due to their respective positive edges to $S$ and $T$.

3. $E^-_M = \bigcup_{v \in V} \{\{M_v, L\}, \{M_v, R\}\}$
   These edges ensure that for all vertices $v \in V$ the interval $I_{M_v}$ lies in $I_M$. See Lemma A.3.

4. $E^-_* = \{\{X_v, M\} : v \in V\}$
   These edges ensure that for all vertices $v \in V$, the interval $I_{X_v}$ either lies in $I_L$ or in $I_R$. See Lemma A.4.

5. $E^-_V = \{\{X_u, X_v\} : u, v \in V, u \neq v\}$
   These edges ensure that for all vertices $u, v \in V$ with $u \neq v$ the intervals $I_{X_u}$ and $I_{X_v}$ are disjoint.

6. $E^-_E = \bigcup_{(u,v) \in E} \{\{X_v, A_u\}, \{X_v, B_u\}\}$
   These edges enforce topological orderings of the vertices. See Lemma A.5.

This concludes the construction. We refer to Figure 5 for an illustration. It is clear that $V'$, $E^+$, and $E^-$ have the sizes claimed in the theorem.

## A.2  Structural lemmas

To prove the correctness of the reduction, we will make use of a few smaller results that describe the structure of any conflict-free interval representation of $G$. First, notice that $\{S, T\} \in E^-$, so for any conflict-free interval representation, it must hold that $I_S \cap I_T = \emptyset$. For the rest of this analysis, assume without loss of generality $r(I_S) < \ell(I_T)$.

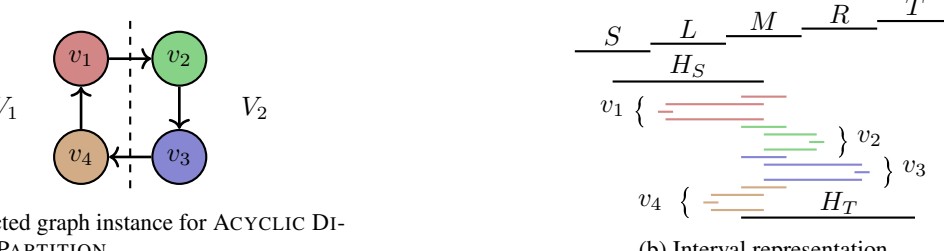

(a) Directed graph instance for ACYCLIC DI-
GRAPH PARTITION

(b) Interval representation

Figure 5: Reduction from ACYCLIC DIGRAPH PARTITION

**Lemma A.1.** *For any conflict-free interval representation of $G$ it must hold that for all $u \in V' \setminus \{S\}$ : $r(I_S) < r(I_u)$ and for all $u \in V' \setminus \{T\} : \ell(I_u) < \ell(I_T)$.*

*Proof.* Towards a contradiction assume there exists a vertex $u \in V' \setminus \{S\}$ such that $r(I_u) \leq r(I_S)$. As $(V', E^+)$ is a cycle, there exists a path from $u$ to $T$ that does not include $S$. The union $I_p$ of the intervals corresponding to the vertices in this path must form an interval itself. As $r(I_u) \leq r(I_S)$, but $\ell(I_T) > r(I_S)$, $I_S \cap I_p \neq \emptyset$, and consequently there exists some vertex $x \in V' \setminus \{S\}, x \neq u$ along this path such that $I_S \cap I_x \neq \emptyset$. Note that by the construction of this path, $x$ cannot be a positive neighbor of $S$, as we explicitly choose one of the two paths from $u$ to $T$ that does *not* include $S$. Then, $\{S, x\} \in E^-$, leading to a contradiction. Finally, for all $u \in V' \setminus \{T\} : \ell(I_u) < \ell(I_T)$ holds by a symmetric argument.

$\square$

**Lemma A.2.** *For any conflict-free interval representation of $G$ it must hold that (i) for all $u \in V' \setminus \{S, L, H_S\} : r(I_S) < \ell(I_u)$ and (ii) for all $u \in V' \setminus \{T, R, H_T\} : r(I_u) < \ell(I_T)$.*

*Proof.* Assume towards a contradiction that there exists a vertex $u \in V' \setminus \{S, L, H_S\}$ such that $\ell(I_u) \leq r(I_S)$. From Lemma A.1, we know $r(I_u) > r(I_S)$, so it follows that $I_u \cap I_S \neq \emptyset$. However, since $\{S, u\} \in E^-$ this leads to a contradiction with the assumption of a conflict-free interval representation. The proof for (ii) follows symmetrically. $\square$

**Lemma A.3.** *For any conflict-free interval representation of $G$, it must hold that for all $v \in V$: $I_{M_v} \subset I_M$.*

*Proof.* By construction, the open interval $(r(I_S), \ell(I_T)) \subset I_L \cup I_M \cup I_R$. From Lemma A.2, we know that $I_{M_v} \subset I_L \cup I_M \cup I_R$. Finally, as $\{M_v, L\}, \{M_v, R\} \in E^-$, the claim holds. $\square$

**Lemma A.4.** *For any conflict-free interval representation of $G$, it must hold that for all $v \in V$ either $I_{X_v} \subset I_L$ or $I_{X_v} \subset I_R$ but not both.*

*Proof.* By construction, the open interval $(r(I_S), \ell(I_T)) \subset I_L \cup I_M \cup I_R$. From Lemma A.2, we know that $I_{X_v} \subset I_L \cup I_M \cup I_R$. Furthermore, we know that $I_L \cap I_R = \emptyset$. Finally, as $\{X_v, M\} \in E^-$ the claim holds. $\square$

Building on this statement, we can further characterize the relative location of the intervals $I_{X_v}$ inside $I_L$ and $I_R$.

**Lemma A.5.** *For any conflict-free interval representation of $G$ it must hold that for all edges $(u, v) \in E$ if $I_{X_u} \subset I_R$ and $I_{X_v} \subset I_R$, then $r(I_{X_u}) < \ell(I_{X_v})$, and, symmetrically, if $I_{X_u} \subset I_L$ and $I_{X_v} \subset I_L$, then $r(I_{X_v}) < \ell(I_{X_u})$*

*Proof.* As we assumed that $r(I_S) < \ell(I_T)$, it follows that $\ell(I_M) < \ell(I_R) \leq r(I_M) < r(I_R)$. Towards a contradiction, assume there exists an edge $(u, v) \in E$ such that $I_{X_u} \subset I_R$ and $I_{X_v} \subset I_R$, but $\ell(I_{X_v}) \leq r(I_{X_u})$. As in any conflict-free interval representation the intervals $I_{X_u}$ and $I_{X_v}$ are disjoint, this implies that $r(I_{X_v}) < \ell(I_{X_u})$. Now, consider the intervals $I_{M_u}$ and $I_{A_u}$. From Lemma A.3, we know that $I_{M_u} \subset I_M$, hence $r(I_{M_u}) < \ell(I_{X_v}) < r(I_{X_v}) < \ell(I_{X_u})$. By

construction, $I_{A_u}$ must overlap with $I_{M_u}$ and $I_{X_u}$, hence $\ell(I_{A_u}) \leq r(I_{M_u})$ and $r(I_{A_u}) \geq \ell(I_{X_u})$. However, this implies that $I_{A_u} \cap I_{X_v} \neq \emptyset$, leading to a violation of the $\{X_v, A_u\}$ constraint introduced in $E_E^-$. By symmetry, this also proves the case where $I_{X_u} \subset I_L$ and $I_{X_v} \subset I_L$. $\qquad\square$

## A.3 Proof of Theorem 2.2

Equipped with Lemmas A.1 to A.5 we can now prove Theorem 2.2.

We first show that if $H = (V, E)$ is a YES-instance of ACYCLIC DIGRAPH PARTITION, then the constructed signed graph instance $G$ has a conflict-free interval-representation in BEST INTERVAL APPROXIMATION. Assume $H[V_1]$ and $H[V_2]$ are the two acyclic induced subgraphs of $H$ corresponding to the partition and let $k = |V_1|$. Further, let $[v_{(1,1)}, \ldots, v_{(1,k)}]$ and $[v_{(2,1)}, \ldots, v_{(2,n-k)}]$ be topological orderings of $V_1$ and $V_2$, respectively. Now, we define intervals for $V_c'$ as follows and depicted in Figure 5:

$I_S := [0, 0.2]$, $I_L := [0.2, 0.4]$, $I_M := [0.4, 0.6]$, $I_R := [0.6, 0.8]$, $I_T := [0.8, 1]$, $I_{H_S} := [0.1, 0.5]$, $I_{H_T} := [0.5, 0.9]$.

This satisfies the constraints set by $E_c^-$. Then, for each $v \in V$, we assign $I_{M_v} = [0.45, 0.55]$. This satisfies all constraints imposed by $E_M^-$. Next, we define

$$\text{for all } i \in \{1, \ldots, k\} : I_{X_{v_{(1,i)}}} = \left[0.4 - \frac{2i+1}{16k}, 0.4 - \frac{2i}{16k}\right], \text{ and}$$

$$\text{for all } i \in \{1, \ldots, n-k\} : I_{X_{v_{(2,i)}}} = \left[0.6 + \frac{2i}{16(n-k)}, 0.6 + \frac{2i+1}{16(n-k)}\right].$$

This ensures that for all vertices $u \in V_1$, the interval $I_{X_u}$ lies in $(0.2, 0.4)$, and symmetrically for all vertices $v \in V_2$, the interval $I_{X_v}$ lies in $(0.6, 0.8)$, hence satisfying $E_*^-$. Further, for all vertices $u, v \in V$ with $u \neq v$ their intervals $I_{X_u}, I_{X_v}$ are disjoint, thereby satisfying $E_V^-$. To conclude the construction of the interval representation, we set

$$\text{for all } v \in V_1 : I_{A_v} = I_{B_v} = [\ell(I_{X_v}), 0.5], \text{ and}$$
$$\text{for all } v \in V_2 : I_{A_v} = I_{B_v} = [0.5, r(I_{X_v})].$$

Now, all the constraints set in $E^+$, $E_S^-$ and $E_T^-$ are satisfied by construction. It is left to check whether the constraints set by $E_E^-$ are satisfied. Here, $I_{X_v}$ must not overlap $I_{A_u}$ or $I_{B_u}$ if there exists a directed edge $(u, v) \in E$. This is trivially satisfied if $u \in V_1$ and $v \in V_2$ or vice-versa. If both $u, v \in V_2$, then in the constructed interval representation we must have that $\ell(I_{X_v}) > r(I_{A_u}) = r(I_{B_u}) = r(I_{X_u})$. As the intervals $\{I_{X_v} : v \in V_2\}$ were constructed according to a topological ordering of $V_2$, this is always satisfied. The argument works symmetrically for $V_1$, and hence the interval representation is conflict-free.

Conversely, suppose the constructed instance $G$ admits a conflict-free interval representation in BEST INTERVAL APPROXIMATION. We claim that this implies $H$ is a YES-instance of ACYCLIC DIGRAPH PARTITION. First, by Lemmas A.1 and A.2, any conflict-free interval representation places $I_S$ and $I_T$ at the extreme left and extreme right, respectively. Consequently, in the open interval $(r(I_S), \ell(I_T))$, the intervals $I_L, I_M$, and $I_R$ appear in that left-to-right order. Next, Lemma A.3 guarantees that every interval $I_{M_v}$ for $v \in V$ is contained in $I_M$. Meanwhile, Lemma A.4 ensures that each $I_{X_v}$ is contained entirely in either $I_L$ or $I_R$. This setup naturally suggests a bipartition of the set $V$:

$$V_1 = \{v \in V : I_{X_v} \subset I_L\} \quad \text{and} \quad V_2 = \{v \in V : I_{X_v} \subset I_R\}.$$

We claim that $H[V_1]$ and $H[V_2]$ must each be acyclic. Indeed, in Lemma A.5 we show that for any directed edge $(u, v) \in E$ with $u, v \in V_1$, the intervals $I_{X_u}$ and $I_{X_v}$ in $I_L$ must satisfy $r(I_{X_v}) < \ell(I_{X_u})$. Hence, the interval $I_{X_v}$ must lie to the left of $I_{X_u}$. This implies a topological ordering of vertices in $V_1$, and thus prevents directed cycles in $H[V_1]$. A symmetric argument shows that $H[V_2]$ is acyclic. Thus, $H$ admits a partition of its vertex set into two DAGs $H[V_1]$ and $H[V_2]$. Therefore, $H$ is a YES-instance of ACYCLIC DIGRAPH PARTITION, completing the proof of Theorem 2.2.

# B A PTAS for a Fixed Number of Intervals

We prove Theorem 2.3, which generalizes a result of Giotis and Guruswami [28] for the agreement version of CORRELATION CLUSTERING in complete signed graphs and also for a fixed number of clusters. Our analysis is similar to that of [28], but we have to adjust it such that we take into account the overlap of the given intervals.

Interestingly, Giotis and Guruswami [28] also presented a PTAS for the disagreement version of CORRELATION CLUSTERING. However, their result does not extend to BEST INTERVAL AP-PROXIMATION, since in our setting we may have overlapping intervals and this breaks their greedy assignment rule, as well as several of their technical arguments.

In the following, we consider a complete signed graph $G = (V, E^+ \cup E^-)$. We assume that we are given $k$ intervals $I_1, \ldots, I_k \subset \mathbb{R}$ as input. In the problem we consider, each vertex must be assigned to one of the intervals such that agreement is maximized.

The main work of this section will go into the proof of the following proposition from which the rest of our results follow.

**Proposition B.1.** *Let $\varepsilon > 0$ and $\delta > 0$. With probability at least $1 - \delta$, Algorithm 2 computes an approximate solution with additive error at most $\varepsilon n^2/2$ and has running time $k^{O(1/\varepsilon^3 \log(k/(\varepsilon\delta)))} \cdot n$.*

The proposition allows us to obtain Corollary B.2, which shows that we can obtain a multiplicative $(1+\varepsilon)$-approximation guarantee for $k$ fixed intervals. This then also implies the proof of Theorem 2.3.

**Corollary B.2.** *Let $\varepsilon > 0$ and $\delta > 0$. There exists an algorithm that computes a $(1+\varepsilon)$-approximate solution for BEST INTERVAL APPROXIMATION with $k$ given intervals in time $k^{O(1/\varepsilon^3 \log(k/(\varepsilon\delta)))} \cdot n$ with probability at least $1 - \delta$.*

*Proof.* First, assume that all pairs of intervals overlap. Then it does not matter how we assign the vertices because only the positive edges can be satisfied. In this case, any assignment will achieve the same objective function value as OPT. Second, assume that there is at least one pair of non-overlapping intervals. In that case, we know from CORRELATION CLUSTERING that the objective function agreement must be $\Omega(n^2)$ (see the proof of Theorem 3.1 in [28]). In that case, we can use the result from Proposition B.1 to obtain a multiplicative $(1 + \varepsilon)$-approximation by making the parameter $\varepsilon$ in our additive approximation small enough. The running time claim follows from Proposition B.1. $\square$

*Proof of Theorem 2.3.* Note that we only need to consider at most $2^{\binom{k}{2}}$ choices for picking $k$ intervals (up to changing their coordinates): For any pair of intervals they either overlap or they do not. Thus, since there are $\binom{k}{2}$ interval pairs, there are at most $2^{\binom{k}{2}} = 2^{O(k^2)}$ choices for the overlap.

Now we can just enumerate all possible overlap-patterns of $k$ intervals and run the algorithm from Corollary B.2 for it in time $2^{O(k^2 + 1/\varepsilon^3 \log(k) \log(k/(\varepsilon\delta)))} \cdot n \leq 2^{O(k^2 \log(k/(\varepsilon\delta))/\varepsilon^3)} \cdot n$. Note that our result is correct if the algorithm succeeds for the $k$ intervals picked by OPT, and thus we get the desired success probability (and in particular we do not have to apply a union bound that the algorithm succeeds for all possible choices of $2^{O(k^2)}$ choices of intervals). $\square$

For the remainder of this section, we work on the proof of Proposition B.1.

We present the pseudocode of our method with full details in Algorithm 2. On a high level, our algorithm works by partitioning $V$ into $m = O(1/\varepsilon)$ equally-sized subsets $V_1, \ldots, V_m$. Then, for each $i = 1, \ldots, m$, we proceed as follows. We sample a set of vertices $S_i \subseteq V \setminus V_i$ of size $\tilde{O}\left(1/\varepsilon^2\right)$. Now, we enumerate all possible assignments of $S_i$ into $(S_{i,1}, \ldots, S_{i,k})$, where $S_{i,\ell} \subseteq S_i$ are the vertices assigned to interval $I_\ell$, and for each such assignment, we greedily assign the vertices $v \in V_i$ to the interval that maximizes the agreement of $v$'s edges to the clustering of $S_i$ given by $(S_{i,1}, \ldots, S_{i,k})$. This process gives a clustering of $V_i$ and we build the final clustering by merging our solutions for the $V_i$ to obtain a global clustering of $V$.

For a high-level description of the algorithm, see Section 2.2.

---

**Algorithm 2:** Maximizing agreement for fixed $k$

---

**Input:** A complete signed graph $G = (V, E^+ \cup E^-)$, contiguous and non-empty intervals
$\quad\quad I_1, \dots, I_k \subset \mathbb{R}, \varepsilon > 0$

**Result:** A clustering $(\mathsf{ALG}_1, \dots, \mathsf{ALG}_k)$ maximizing the agreement

**1** Partition $V$ into $m = \frac{4}{\varepsilon}$ sets $V_1, \dots, V_m$ of size $\frac{n}{m} = \frac{\varepsilon n}{4}$ each;

**2** Sample $S_i \subseteq V \setminus V_i$ uniformly at random with replacement of size $s = \frac{32^2}{2\varepsilon^2} \log \left( \frac{64mk}{\varepsilon\delta} \right)$ for all
$\quad\quad i = 1, \dots, m$;

**3** Initialize some arbitrary clustering $(\mathsf{ALG}_1, \dots, \mathsf{ALG}_k)$;

**4 for** *all possible clusterings of all $S_i$ into $(S_{i,1}, \dots, S_{i,k})$* **do**

**5** $\quad$ **for** $i = 1, \dots, m$ **do**

**6** $\quad\quad$ Let $\mathsf{ALG}'_{i,1}, \dots, \mathsf{ALG}'_{i,k}$ be an empty clustering of $V_i$;

**7** $\quad\quad$ **for** $u \in V_i$ **do**

**8** $\quad\quad\quad$ $\ell^* \leftarrow \mathrm{argmax}_{\ell=1,\dots,k} \, \mathsf{agree}(u, \ell, (S_{i,1}, \dots, S_{i,k}))$;

**9** $\quad\quad\quad$ Assign $u$ to $\mathsf{ALG}_{i,\ell^*}$;

**10** $\quad$ Set $\mathsf{ALG}'_\ell \leftarrow \bigcup_{i=1}^m \mathsf{ALG}'_{i,\ell}$ for all $\ell = 1, \dots, k$;

**11** $\quad$ **if** *agree*$(G, (\mathsf{ALG}'_1, \dots, \mathsf{ALG}'_k)) >$ *agree*$(G, (\mathsf{ALG}_1, \dots, \mathsf{ALG}_k))$ **then**

**12** $\quad\quad$ Set $(\mathsf{ALG}_1, \dots, \mathsf{ALG}_k) \leftarrow (\mathsf{ALG}'_1, \dots, \mathsf{ALG}'_k)$;

**13 return** $(\mathsf{ALG}_1, \dots, \mathsf{ALG}_k)$;

---

Recall from the main text that we write $\mathsf{overlap}(\ell)$ to denote the set of all intervals $I_{\ell'}$ that overlap with interval $I_\ell$, i.e., $\mathsf{overlap}(\ell) = \{\ell' : I_\ell \cap I_{\ell'} \neq \emptyset\}$. Furthermore, a *clustering* $C_1, \dots, C_k$ of $V$ is an assignment of the vertices $V$ to intervals. In particular, $C_i$ denotes all vertices which are assigned to interval $I_i$. Note that the $C_i$ are mutually disjoint. For a vertex $u$ and a clustering $C_1, \dots, C_k$ we write

$$\mathsf{agree}(u, \ell, (C_1, \dots, C_k)) = \sum_{\ell' \in \mathsf{overlap}(\ell)} \left| N^+(u) \cap C_{\ell'} \right| + \sum_{\ell' \notin \mathsf{overlap}(\ell)} \left| N^-(u) \cap C_{\ell'} \right|$$

which is the number of agreeing edges of vertex $u$ for the clustering $C_1, \dots, C_k$ when assigning $u$ to interval $I_\ell$. Similarly, we define

$$\mathsf{agree}^+(u, \ell, (C_1, \dots, C_k)) = \sum_{\ell' \in \mathsf{overlap}(\ell)} \left| N^+(u) \cap C_{\ell'} \right|,$$

and

$$\mathsf{agree}^-(u, \ell, (C_1, \dots, C_k)) = \sum_{\ell' \notin \mathsf{overlap}(\ell)} \left| N^-(u) \cap C_{\ell'} \right|.$$

For the analysis we consider an optimal clustering denoted by $\mathsf{OPT} = (\mathsf{OPT}_1, \dots, \mathsf{OPT}_k)$. Here, $\mathsf{OPT}_\ell \subseteq V$ consists of all vertices that get assigned to interval $I_\ell$ in the optimal solution.

We set $\mathsf{OPT}_{i,\ell} = V_i \cap \mathsf{OPT}_\ell$. Note that $\mathsf{OPT}^i = (\mathsf{OPT}_{i,1}, \dots, \mathsf{OPT}_{i,k})$ is the clustering of $\mathsf{OPT}$ when constrained on the vertices in $V_i$.

Next, we construct a set of *hybrid* clusterings that use a part of our solution from $\mathsf{ALG}$ and a part of the solution from $\mathsf{OPT}$. In particular, we set

$$H_{i,\ell} = \left( \bigcup_{j=1}^{i-1} \mathsf{ALG}_{j,\ell} \right) \cup \left( \bigcup_{j=i}^{m} \mathsf{OPT}_{j,\ell} \right).$$

Note that $H_{i,\ell}$ corresponds to a hybrid between $\mathsf{ALG}_\ell$ and $\mathsf{OPT}_\ell$ where all vertices in $V_1, \dots, V_{i-1}$ are clustered based on $\mathsf{ALG}_\ell$ and all vertices in $V_i, \dots, V_m$ are clustered based on $\mathsf{OPT}_\ell$.

Now, additionally we set $\mathcal{H}_{i,\ell} = H_{i,\ell} \setminus V_i$, i.e., these are all vertices in $H_{i,\ell}$ which are not contained in $V_i$ and thus they will be clustered either before or after the $i$'th iteration of our algorithm. We set $H_i = (H_{i,1}, \dots, H_{i,k})$ and note that $H_i$ is a solution for all vertices (not just the vertices in $V_i$). In particular, note that $H_1 = \mathsf{OPT}$ and $H_{m+1} = \mathsf{ALG}$. Additionally, we set $\mathcal{H}_i = (\mathcal{H}_{i,1}, \dots, \mathcal{H}_{i,k})$ for the clustering given by $H_i$ after removing the vertices in $V_i$.

For the rest of the analysis, for all $i = 1, \ldots, m$, we consider the clustering $S_{i,1}, \ldots, S_{i,k}$ of $S_i$ that agrees with the hybrid clustering, i.e., we assume that $S_{i,\ell} = S_i \cap \mathcal{H}_{i,\ell}$. Note that this clustering must be considered by the algorithm since we exhaustively enumerate all possible clusterings of all $S_i$.

**Lemma B.3.** *Let $i \in \{1, \ldots, m\}$. With probability at least $1 - \frac{\delta}{4m}$ over the randomness in $S_i$ the following event happens: For at least a $(1 - \varepsilon/8)$-fraction of the vertices $u \in V_i$ it holds that for all $\ell = 1, \ldots, k$,*

$$\left| \frac{|V \setminus V_i|}{s} \mathsf{agree}^+(u, \ell, (S_{i,1}, \ldots, S_{i,k})) - \mathsf{agree}^+(u, \ell, (\mathcal{H}_{i,1}, \ldots, \mathcal{H}_{i,k})) \right| \leq \frac{\varepsilon}{32} |V \setminus V_i|. \quad (2)$$

*Proof.* Consider any $u \in V_i$ and let $S_i = \{v_1, \ldots, v_s\}$. For $\ell = 1, \ldots, k$ and $j = 1, \ldots, s$, let $X_{j,\ell}$ be the indicator random variable which is 1 if $v_j \in N^+(u)$ and $v_j \in S_{i,\ell'}$ for some cluster with $\ell' \in \mathsf{overlap}(\ell)$, and 0 otherwise.

Note that $\sum_{j=1}^s X_{j,\ell} = \mathsf{agree}^+(u, \ell, (S_{i,1}, \ldots, S_{i,k}))$ and that the $X_{j,\ell}$ are i.i.d. random variables with expectation $\mathbb{E}[X_{j,\ell}] = \mathbf{Pr}(X_{j,\ell} = 1) = \frac{\mathsf{agree}^+(u,\ell,(\mathcal{H}_{i,1},\ldots,\mathcal{H}_{i,k}))}{|V \setminus V_i|}$. Now an additive Chernoff bound gives that

$$\mathbf{Pr}\left( \left| \frac{\mathsf{agree}^+(u, \ell, (S_{i,1}, \ldots, S_{i,k}))}{s} - \frac{\mathsf{agree}^+(u, \ell, (\mathcal{H}_{i,1}, \ldots, \mathcal{H}_{i,k}))}{|V \setminus V_i|} \right| > \frac{\varepsilon}{32} \right)$$
$$< 2 \exp\left( -2 \left( \frac{\varepsilon}{32} \right)^2 s \right) < \frac{\varepsilon \delta}{32mk}.$$

Note that this gives us the inequality from the lemma after multiplying with $|V \setminus V_i|$ on both sides.

Now let $Y$ be the random variable denoting the number of vertices in $V_i$ which do not satisfy the inequality above. Observe that $\mathbb{E}[Y] < \frac{\varepsilon \delta}{32mk} |V_i|$. By Markov's inequality, we get that the inequality holds for all but $\frac{\varepsilon}{8} |V_i|$ vertices with probability at least $1 - \frac{\delta}{4mk}$.

Now the lemma follows by applying a union bound. $\square$

We note that the lemma also holds for $\mathsf{agree}^-(u, \ell, (S_{i,1}, \ldots, S_{i,k}))$ with the same proof.

**Lemma B.4.** *For $i = 0, \ldots, m$ it holds that $\mathsf{agree}(G, H_{i+1}) \geq \mathsf{agree}(G, \mathsf{OPT}) - i \cdot \frac{1}{8} \varepsilon^2 n^2$.*

*Proof.* Consider some iteration $i$ of the algorithm. Note that in this iteration only the vertices in $V_i$ are assigned to clusters and thus $H_i$ and $H_{i+1}$ only differ by the vertices contained in $V_i$. Therefore, our proof will proceed by considering a vertex $u \in V_i$ that gets assigned to interval $I_\ell$ by the algorithm but to interval $I_{\ell'}$ in the solution $H_i$.

First, observe that since the algorithm assigned $u$ to interval $I_\ell$ we must have that

$$\mathsf{agree}^+(u, \ell, (S_{i,1}, \ldots, S_{i,k})) + \mathsf{agree}^-(u, \ell, (S_{i,1}, \ldots, S_{i,k}))$$
$$\geq \mathsf{agree}^+(u, \ell', (S_{i,1}, \ldots, S_{i,k})) + \mathsf{agree}^-(u, \ell', (S_{i,1}, \ldots, S_{i,k}))$$

which implies that

$$\mathsf{agree}^+(u, \ell', (S_{i,1}, \ldots, S_{i,k})) + \mathsf{agree}^-(u, \ell', (S_{i,1}, \ldots, S_{i,k}))$$
$$- \mathsf{agree}^+(u, \ell, (S_{i,1}, \ldots, S_{i,k})) - \mathsf{agree}^-(u, \ell, (S_{i,1}, \ldots, S_{i,k})) \leq 0.$$

Now set $\alpha_i = \frac{|V \setminus V_i|}{s}$ and assume that $u$ is a vertex satisfying Equation (2). Observe that the number of agreements we might lose by this misplacement is at most

$$\mathsf{agree}(u, \ell', (\mathcal{H}_{i,1}, ..., \mathcal{H}_{i,k})) - \mathsf{agree}(u, \ell, (\mathcal{H}_{i,1}, ..., \mathcal{H}_{i,k}))$$

$$= \mathsf{agree}^+(u, \ell', (\mathcal{H}_{i,1}, ..., \mathcal{H}_{i,k})) + \mathsf{agree}^-(u, \ell', (\mathcal{H}_{i,1}, ..., \mathcal{H}_{i,k}))$$
$$\quad - \mathsf{agree}^+(u, \ell, (\mathcal{H}_{i,1}, ..., \mathcal{H}_{i,k})) - \mathsf{agree}^-(u, \ell, (\mathcal{H}_{i,1}, ..., \mathcal{H}_{i,k}))$$

$$= \mathsf{agree}^+(u, \ell', (\mathcal{H}_{i,1}, ..., \mathcal{H}_{i,k})) + \alpha_i \mathsf{agree}^+(u, \ell', (S_{i,1}, ..., S_{i,k})) - \alpha_i \mathsf{agree}^+(u, \ell', (S_{i,1}, ..., S_{i,k}))$$
$$\quad + \mathsf{agree}^-(u, \ell', (\mathcal{H}_{i,1}, ..., \mathcal{H}_{i,k})) + \alpha_i \mathsf{agree}^-(u, \ell', (S_{i,1}, ..., S_{i,k})) - \alpha_i \mathsf{agree}^-(u, \ell', (S_{i,1}, ..., S_{i,k}))$$
$$\quad - \mathsf{agree}^+(u, \ell, (\mathcal{H}_{i,1}, ..., \mathcal{H}_{i,k})) + \alpha_i \mathsf{agree}^+(u, \ell, (S_{i,1}, ..., S_{i,k})) - \alpha_i \mathsf{agree}^+(u, \ell, (S_{i,1}, ..., S_{i,k}))$$
$$\quad - \mathsf{agree}^-(u, \ell, (\mathcal{H}_{i,1}, ..., \mathcal{H}_{i,k})) + \alpha_i \mathsf{agree}^-(u, \ell, (S_{i,1}, ..., S_{i,k})) - \alpha_i \mathsf{agree}^-(u, \ell, (S_{i,1}, ..., S_{i,k}))$$

$$\leq \left| \mathsf{agree}^+(u, \ell', (\mathcal{H}_{i,1}, ..., \mathcal{H}_{i,k})) - \alpha_i \mathsf{agree}^+(u, \ell', (S_{i,1}, ..., S_{i,k})) \right|$$
$$\quad + \left| \mathsf{agree}^-(u, \ell', (\mathcal{H}_{i,1}, ..., \mathcal{H}_{i,k})) - \alpha_i \mathsf{agree}^-(u, \ell', (S_{i,1}, ..., S_{i,k})) \right|$$
$$\quad + \left| \mathsf{agree}^+(u, \ell, (\mathcal{H}_{i,1}, ..., \mathcal{H}_{i,k})) - \alpha_i \mathsf{agree}^+(u, \ell, (S_{i,1}, ..., S_{i,k})) \right|$$
$$\quad + \left| \mathsf{agree}^-(u, \ell, (\mathcal{H}_{i,1}, ..., \mathcal{H}_{i,k})) - \alpha_i \mathsf{agree}^-(u, \ell, (S_{i,1}, ..., S_{i,k})) \right|$$
$$\quad + \alpha_i \mathsf{agree}^+(u, \ell', (S_{i,1}, ..., S_{i,k})) + \alpha_i \mathsf{agree}^-(u, \ell', (S_{i,1}, ..., S_{i,k}))$$
$$\quad - \alpha_i \mathsf{agree}^+(u, \ell, (S_{i,1}, ..., S_{i,k})) - \alpha_i \mathsf{agree}^-(u, \ell, (S_{i,1}, ..., S_{i,k}))$$

$$\leq 4 \cdot \frac{\varepsilon}{32} |V \setminus V_i| + 0$$

$$\leq \frac{\varepsilon}{8} n.$$

Thus we get that for all vertices in $V_i$ satisfying Equation (2) we get that their total difference is at most $\frac{\varepsilon}{8} n |V_i| = \frac{\varepsilon^2 n^2}{32}$.

Furthermore, as there are at most $\frac{\varepsilon}{8} |V_i|$ vertices that do not satisfy Equation (2), they can contribute at most $\frac{\varepsilon}{8} |V_i| n = \frac{\varepsilon^2 n^2}{32}$ edges that are in disagreement. The number of disagreements within $V_i$ is at most $|V_i|^2 = \frac{\varepsilon^2 n^2}{16}$.

In total, we get that we have introduced $\frac{\varepsilon^2 n^2}{8}$ new disagreements due to our approximations in the $i$'th iteration. The lemma now follows by induction. $\square$

*Proof of Proposition B.1.* The approximation ratio follows from the previous lemma. The running time follows from the fact that there are $k^s = k^{O(1/\varepsilon^2 \log(k/(\varepsilon\delta)))}$ choices to assign the $s$ vertices in each $S_i$ to the $k$ intervals and we have to consider the combinations of these assignments for $m = O(1/\varepsilon)$ sets. Thus, the outer loop iterates over $k^{ms} = k^{O(1/\varepsilon^3 \log(k/(\varepsilon\delta)))}$ assignments. Furthermore, each loop iteration can be implemented in time $O(ns)$. $\square$

## C Additional Experiment Details and Results

### C.1 Experiment setup

*Hardware.* All our algorithms are implemented in the Rust programming language. The experiments were run on a system with two AMD EPYC 9124 CPUs, 500 GB of RAM, and an NVIDIA RTX 4000 Ada Generation GPU.

*Dataset preprocessing.* As a preprocessing step, we convert all networks into simple undirected graphs by removing loops and multiple edges. When removing multiple edges between two vertices, we replace them with a single undirected edge whose weight is the sum of the original edges. Finally, we apply a thresholding function where positive edges are assigned weight 1, and negative edges are assigned weight $-1$. An overview of all our datasets is shown in Table 2.

### C.2 Parameters and implementation details for `VENUS` and `GAIA`

The pseudocode of `GAIA` is shown in Algorithm 1. The algorithm uses a greedy approach and employs randomness when partially destroying and reconstructing solutions to improve the overall objective.

Table 2: Summary of our datasets.

| Dataset | $|V|$ | $|E|$ | $|E^+|$ | $|E^-|$ |
|---|---|---|---|---|
| BitcoinOTC | 5 881 | 21 434 | 18 281 | 3 153 |
| Chess | 7 301 | 32 650 | 19 046 | 13 604 |
| WikiElec | 7 115 | 100 355 | 78 440 | 21 915 |
| Bundestag | 1 480 | 397 497 | 320 956 | 76 541 |
| Slashdot | 82 140 | 498 532 | 380 933 | 117 599 |
| Epinions | 131 580 | 708 507 | 589 888 | 118 619 |
| WikiSigned | 138 587 | 712 337 | 628 000 | 84 337 |
| WikiConflict | 116 836 | 2 014 053 | 762 999 | 1 251 054 |

*Batch size.* After one initial assignment performed over the full vertex set, $V$ is partitioned into sets of size $\frac{n}{m}$. Sequentially, each of these $m$ subsets is unassigned and reassigned greedily. In the presented results, we generally use $m = 10$ for the sake of consistency. Depending on the specific dataset, higher or lower values might be advantageous. Lower numbers result in more aggressive reassignments and higher numbers in smaller, more incremental changes. For example, for BitcoinOTC, we observed that solutions with $m = 100$ on average achieve 1% better objective than with $m = 10$.

*Breaking ties and randomization.* An essential aspect of the implementation is randomised tie breaking for both the order of vertex assignment *and* the selection of the assigned cluster. Without the extensive use of randomization, the solution quality suffers greatly, and GAIA can get stuck in bad local optima early. The same can be said for the ordering of the reassignment: if vertices are reassigned in the same order in every iteration, the performance of the algorithms is substantially reduced. In our implementation, vertices are greedily assigned during the first epoch, and in each subsequent epoch, they are randomly partitioned into batches.

*Vertex priority.* In each batch of vertices, we select the assignment order based on the maximal potential agreement of the vertex. In our preliminary experiments performed during the algorithm development, this strategy was more effective than using a randomized order or using different characteristics like the vertex degree for ordering. However, the performance does not differ greatly, so other methods of selecting vertex ordering might be considered in the future.

*Timeout and early stopping.* In the presented results, both GAIA and VENUS terminate either after 30 minutes by timeout, or until the best found solution could not be improved for 50 consecutive epochs. The condition for early stopping is intentionally chosen in a very conservative manner, as often the algorithms will find very marginal improvements to their solutions late in the optimization process. To reduce runtime, using, e.g., a time limit of 3 minutes instead of 30 massively reduces solving time while only causing a significant decrease in objective value for the Epinions and WikiConflict dataset, with an average relative difference of 1% and 10% respectively.

*Simulated annealing for VENUS.* With the implementation of GAIA, even with the added randomness, a high variance in the solution quality can be observed. This, paired with the fast convergence time, motivated a second approach with simulated annealing: VENUS. For VENUS, every part of the algorithm remains unchanged, besides the assignment process in Line 7 of Algorithm 1. We replace the deterministic argmax with a temperature-scaled softmax over the agreement, where we define softmax over some function $f$ as

$$\text{softmax}_{\ell=1\ldots k} f(\ell) = \frac{\exp(f(\ell))}{\sum_{\ell'=1\ldots k} \exp(f(\ell'))}. \tag{3}$$

For our setting, we use the agreement scaled by a temperature parameter $t$ to control the level of randomness, with higher levels of $t$ yielding close to uniformly random distributions, and small values of $t$ resulting in increasingly greedy solutions. In all our experiments, we use an initial temperature of $t_0 = 100$, and use an exponential decay schedule where after every $\tau = 5$ epochs we let $t \leftarrow t \cdot \alpha$, where $\alpha = \frac{2}{3}$.

*Interval structure.* In all experimental results presented in the main text, we use 8-Chains as the interval structure for both GAIA and VENUS. As seen in Figures 3b and 6 and Table 3, using larger and thus more expressive chain-like structures did not substantially improve the objective value. For some

Table 3: Comparison of GAIA and VENUS for BEST INTERVAL APPROXIMATION across 4/8/12/16-Chain. For each chain length, we report the best (lowest) and average disagreement. Across all datasets, improvement above 8-Chains is minimal, with WikiConflict being the only dataset with more than 1% improvement.

| Dataset | Algorithm | 4-Chain | | 8-Chain | | 12-Chain | | 16-Chain | |
|---|---|---|---|---|---|---|---|---|---|
| | | Best | Avg | Best | Avg | Best | Avg | Best | Avg |
| BitcoinOTC | GAIA | 768 | 824 | 711 | 767 | 721 | 766 | 725 | 769 |
| | VENUS | 819 | 839 | 760 | 783 | 750 | 787 | 716 | 781 |
| Chess | GAIA | 7 279 | 7 361 | 6 472 | 6 573 | 6 469 | 6 590 | 6 467 | 6 620 |
| | VENUS | 7 276 | 7 334 | 6 410 | 6 486 | 6 426 | 6 505 | 6 398 | 6 516 |
| WikiElec | GAIA | 11 937 | 12 159 | 11 275 | 11 459 | 11 265 | 11 403 | 11 300 | 11 474 |
| | VENUS | 11 923 | 11 943 | 11 297 | 11 420 | 11 302 | 11 404 | 11 255 | 11 390 |
| Bundestag | GAIA | 3 772 | 3 903 | 1 001 | 9 054 | 1 030 | 10 256 | 2 584 | 10 845 |
| | VENUS | 3 819 | 4 324 | 1 001 | 2 078 | 1 002 | 10 862 | 11 704 | 11 722 |
| Slashdot | GAIA | 51 367 | 53 042 | 45 120 | 46 218 | 44 860 | 46 180 | 44 827 | 45 910 |
| | VENUS | 51 117 | 51 813 | 44 563 | 44 884 | 44 447 | 44 767 | 44 462 | 44 707 |
| Epinions | GAIA | 34 213 | 34 628 | 31 687 | 32 789 | 31 959 | 33 125 | 31 805 | 32 755 |
| | VENUS | 34 112 | 34 200 | 31 286 | 33 258 | 31 287 | 32 849 | 31 445 | 34 438 |
| WikiSigned | GAIA | 38 513 | 39 313 | 35 204 | 36 270 | 35 003 | 36 225 | 35 008 | 36 269 |
| | VENUS | 38 083 | 38 343 | 34 564 | 34 971 | 34 819 | 35 230 | 35 062 | 35 390 |
| WikiConflict | GAIA | 140 004 | 142 435 | 69 344 | 69 847 | 68 094 | 68 445 | 67 761 | 68 111 |
| | VENUS | 142 450 | 143 269 | 69 014 | 69 264 | 67 645 | 67 953 | 67 472 | 67 717 |

datasets, like Bundestag or Epinions, performance even starts to decrease with more intervals. Hence, we report the results on an 8-Chain for consistency and to strike a balance between interpretability, expressivity, and effectiveness of our heuristic.

### C.3 Choosing interval structures

We always choose the interval structure for the algorithm statically. While we can adapt partial assignments to changed interval structures, preliminary experiments indicate that pruning unused intervals does not provide an effective heuristic. Rather, we would suggest using approaches from hyperparameter search to choose interval structures. With only very limited runtime, GAIA and VENUS can find good solutions. Methods like successive halving [32] can be used to exploit this efficiency and start with a large population of candidate structures, which is quickly reduced to only invest compute into structures with good objective scores.

### C.4 Impact of simulated annealing on solution quality

Next, we study how much the simulated annealing used in VENUS impacts its results compared to GAIA. Our detailed results on the real-world datasets are shown in Table 4. For almost all datasets, the average objective function of solutions found by VENUS is better than those found by GAIA. This demonstrates how simulated annealing can avoid local minima and continue the optimization process toward better solutions. Additionally, VENUS generally exhibits smaller standard deviations, indicating more consistency in the approach. A clear outlier to this is the Epinions dataset, where the trend reverses and the purely greedy approach performs better; we tried to understand the reason for this behavior, but could not find a clear indicator.

### C.5 Benchmarking against CORRELATION CLUSTERING baselines

In addition to using GAIA and VENUS to find interval assignments with partially overlapping structures, we also evaluate their performance on structures where intervals are non-overlapping (pairwise disjoint). As described in Section 2, this setup corresponds to the CORRELATION CLUSTERING

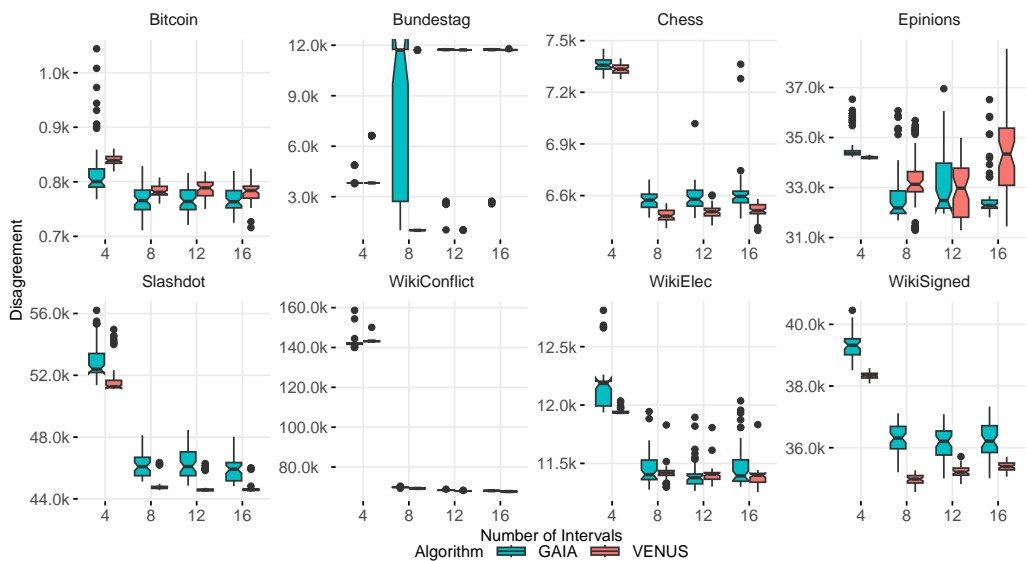

Figure 6: Final objective for different interval structures for all datasets.

Table 4: Comparison of GAIA and VENUS for solving BEST INTERVAL APPROXIMATION. We report the minimum disagreement (in absolute number of edges), as well as averages and standard deviations over 50 runs. The solutions were computed for a fixed 8-Chain interval structure and 10 reassignment batches.

| | Gaia | | Venus | |
|---|---|---|---|---|
| Dataset | Best | Avg±Std | Best | Avg±Std |
| BitcoinOTC | 711 | 767±26 | 760 | 783±11 |
| Chess | 6 472 | 6 573±52 | 6 410 | 6 486±36 |
| WikiElec | 11 275 | 11 459±143 | 11 297 | 11 420±75 |
| Bundestag | 1 001 | 9 054±4 357 | 1 001 | 2 078±3 244 |
| Slashdot | 45 120 | 46 218±860 | 44 563 | 44 884±473 |
| Epinions | 31 687 | 32 789±1 300 | 31 286 | 33 258±1 306 |
| WikiSigned | 35 204 | 36 270±509 | 34 564 | 34 971±165 |
| WikiConflict | 69 344 | 69 847±193 | 69 014 | 69 264±141 |

problem with a fixed number of clusters. Results for a structure with 8 disjoint intervals are shown in Tables 5 and 6. Across all datasets, GAIA and VENUS achieve objective values within 0.5 percentage points of the best CORRELATION CLUSTERING baselines, despite being constrained to only 8 clusters. Moreover, the solutions produced by GAIA and VENUS show low variance, with a standard deviation of under 100 violations on all but one dataset. This indicates that the algorithms consistently found strong solutions, particularly with less variance than when using overlapping interval structures (see Table 4).

We believe that this finding is highly interesting, since it shows that our methods can find CORRELA-TION CLUSTERING solutions that are on par with state-of-the-art algorithms, while also being able to solve our more general BEST INTERVAL APPROXIMATION problem.

Additionally, it is notable that we find competitive solutions even though we only use 8 clusters, whereas the baselines might use an unrestricted number. However, this finding is not completely new and echoes findings by Brusco and Doreian [9], who observed similar objective values using few clusters on the WikiElec and Slashdot datasets.

Table 5: Comparison of our algorithms and CORRELATION CLUSTERING baselines for solving CORRELATION CLUSTERING. Reported is the percentage of edges violated in the solution (lower is better). Our algorithms use a fixed structure of 8 distinct clusters, while the CORRELATION CLUSTERING algorithms are not restricted in the number of clusters.

| | Our algorithms | | CORRELATION CLUSTERING baselines | | | |
|---|---|---|---|---|---|---|
| Dataset | GAIA | VENUS | GAEC | GAECKLj | SCMLEvo | RAMA |
| BitcoinOTC | **5.57** | **5.57** | 5.58 | **5.57** | **5.57** | 5.64 |
| Chess | 27.67 | 27.75 | 28.64 | 28.10 | **27.33** | 39.98 |
| WikiElec | **14.13** | **14.13** | **14.13** | **14.13** | **14.13** | 14.45 |
| Bundestag | **2.95** | **2.95** | 3.06 | **2.95** | **2.95** | 3.72 |
| Slashdot | 13.70 | 13.59 | 13.75 | 13.66 | **13.52** | 17.17 |
| Epinions | 6.69 | 6.71 | 6.83 | 6.68 | **6.67** | 6.86 |
| WikiSigned | 6.30 | 6.21 | **6.17** | **6.17** | **6.17** | 6.96 |
| WikiConflict | 5.85 | 5.85 | 5.87 | **5.82** | **5.82** | 6.02 |

Table 6: Comparison of the results of GAIA and VENUS when solving CORRELATION CLUSTERING. We report the minimum disagreement (in absolute number of edges), as well as averages and standard deviations over 50 runs. The solutions were computed for 8 distinct clusters (i.e., 8 non-overlapping intervals) and 10 reassignment batches.

| | Gaia | | Venus | |
|---|---|---|---|---|
| Dataset | Best | Avg±Std | Best | Avg±Std |
| BitcoinOTC | 1 194 | 1 201±3 | 1 194 | 1 200±4 |
| Chess | 9 035 | 9 128±55 | 9 061 | 9 132±72 |
| WikiElec | 14 181 | 14 185±2 | 14 182 | 14 185±2 |
| Bundestag | 11 736 | 11 756±27 | 11 725 | 11 736±9 |
| Slashdot | 68 276 | 68 351±30 | 67 759 | 68 334±88 |
| Epinions | 47 412 | 47 499±36 | 47 570 | 48 750±434 |
| WikiSigned | 44 881 | 44 992±148 | 44 217 | 44 274±23 |
| WikiConflict | 117 814 | 117 882±23 | 117 885 | 117 932±26 |

## C.6 Scalability and runtime analysis

The time until convergence for GAIA and VENUS is shown in Table 7 and Figure 7. For both algorithms, the running time scales roughly linearly with the size of the graph and the memory usage is at most 260 megabytes for the largest datasets. Despite the artificial slowdown of convergence during early epochs, VENUS is remarkably not notably slower to converge than GAIA. Further, we note that both the memory and runtime complexity of our algorithms depend linearly on the number of intervals, so the running time can vary with the size of the interval structure.

## C.7 Case study

*The* Bundestag *dataset.* Our novel Bundestag dataset was constructed by scraping all the roll-call voting data from the German parliament between October 18th, 2012, and March 18th, 2025.[3] As official works, this voting data is not subject to copyright. From the voting data, we then generated a signed graph by representing each member with a vertex and assigning a positive edge if two members vote the same way in at least 75% of sessions they both attended, and a negative edge if their votes aligned in 25% of sessions or less. This left two politicians without any edges, which were thus excluded from the signed graph. Notably, the voting data includes several politicians who changed parties during their parliamentary careers. In such cases, we treat the politicians as members of the party they were first affiliated with for visualization purposes. This could also explain slight within-party differences, e.g., the LINKE party split between 2023 and 2024.

---

[3]This data is publicly available at https://www.bundestag.de/parlament/plenum/abstimmung/liste

Table 7: Time until convergence averaged over 50 runs on different instances. We report the runtime in seconds until 50 full epochs without improvement (lower is better). Our algorithms use a fixed 8-Chain interval structure and 10 batches for vertex reassignment.

| Dataset | GAIA | | | VENUS | | |
| --- | --- | --- | --- | --- | --- | --- |
| | Best | Avg±Std | Worst | Best | Avg±Std | Worst |
| BitcoinOTC | 1.2 | 2.5±0.9 | 5.7 | 1.8 | 3.2±0.8 | 6.2 |
| Chess | 3.7 | 8.4±2.4 | 15.2 | 3.8 | 8.5±2.1 | 13.4 |
| WikiElec | 6.1 | 16.2±5.5 | 31.1 | 7.1 | 15.9±6.2 | 41.5 |
| Bundestag | 5.5 | 8.8±1.9 | 16.2 | 10.9 | 15.2±2.4 | 22.8 |
| Slashdot | 107.5 | 251.9±114.8 | 781.2 | 98.9 | 173.8±45.3 | 348.5 |
| Epinions | 182.3 | 369.1±129.4 | 954.8 | 196.0 | 390.8±143.0 | 897.4 |
| WikiSigned | 169.6 | 353.5±94.3 | 602.9 | 214.6 | 357.5±90.6 | 585.3 |
| WikiConflict | 436.2 | 874.0±192.8 | 1 470.0 | 372.9 | 695.1±183.7 | 1 220.0 |

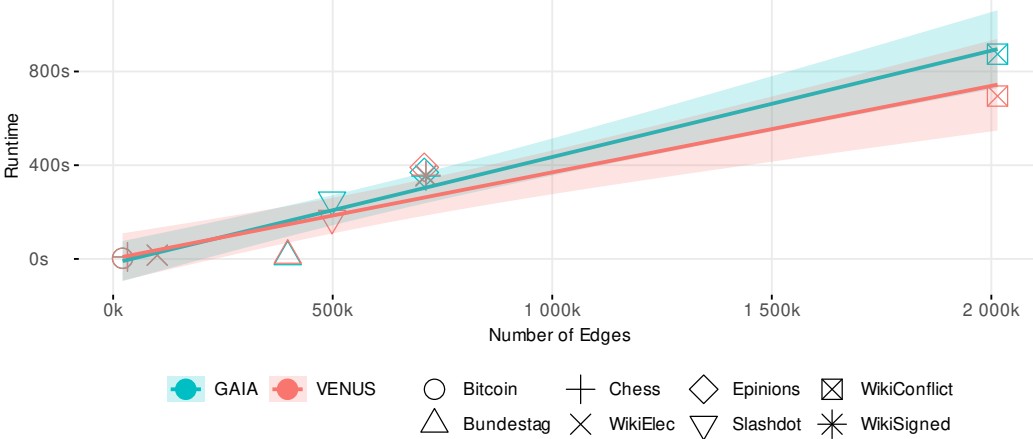

Figure 7: Time until convergence per instance, see Table 7 for details.

*Generation of Figure 1.* With the preprocessed Bundestag instance, we let VENUS compute a solution and for each politician we visualize their party membership via coloring the corresponding point and the positioning it on the y-axis. The interval assignment produced by VENUS determines the x-position. We then slightly adjust the position inside the bins to indicate the affinity of each politician as follows: For the vertex $v \in C_\ell$ corresponding to the politician, we define its *affinity* to other clusters $C_{\ell'}$ as

$$\mathsf{affinity}(v, C_{\ell'}) = \left| N^+(v) \cap C_{\ell'} \right| - \left| N^-(v) \cap C_{\ell'} \right|. \tag{4}$$

Then we calculate the $x$-perturbation of $v$, denoted by $v_x$, as

$$v_x = \tanh \left( \sum_{\ell' \neq \ell} \frac{\mathsf{affinity}(v, C_{\ell'})}{(\ell' - \ell)(|N^+(v)| + |N^-(v)|)} \right). \tag{5}$$

In words, the position of a vertex is shifted to the left or to the right according to the assignment of its neighbors, depending on their distance. Neighbors assigned to closer intervals affect the position more, and neighbors assigned further away affect the position less. The direction of this influence depends on the edge sign.

This function simply serves the purpose of visualization, to show connectivity across intervals. It is not a direct output of our algorithm and does not have a fixed interpretation. Finally, we slightly adjust the position of the vertices in the plot with small amounts of Gaussian noise to avoid over-plotting. The figure also includes a violin plot of each political party.

*Results.* The solution found by VENUS is highly interpretable and can be used to generate insight about the co-voting behavior in the German parliament. This interpretation is discussed in Section 4,

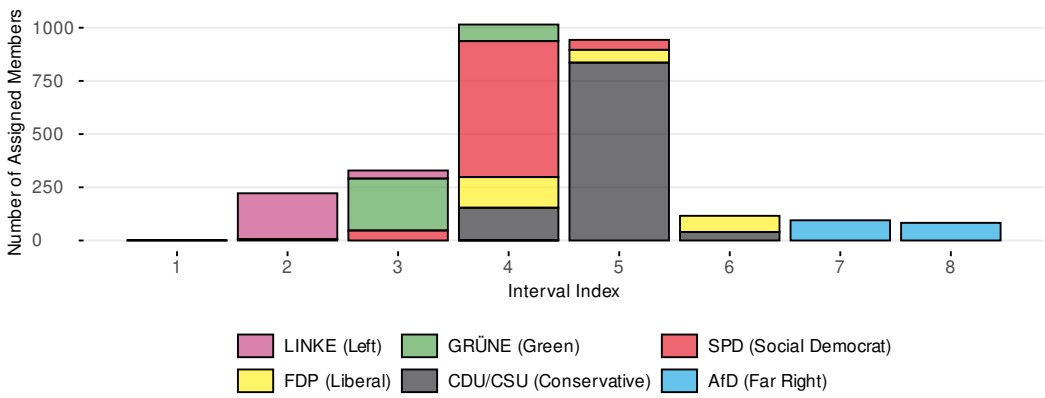

Figure 8: An alternative visualization of VENUS's results on the Bundestag dataset. Here, we present party affiliations of Bundestag members assigned to each interval.

with an alternative visualisation in Figure 8, which shows an interval-centric view instead of the party-centric view in Figure 1. Figure 8 shows that each interval is dominated by members of one political party. Further, as discussed previously, the figure accurately reflects the political spectrum in Germany, with the exception of the FDP, which appears "split" between intervals 4–6. This can be justified by the different coalition structures in the last few legislative periods, as we discussed in the main text.

