# OpenReview forum: "Discovering Opinion Intervals from Conflicts in Signed Graphs"
_NeurIPS.cc/2025/Conference — NeurIPS 2025 oral_

### Official Review · Reviewer_NU77 · 2025-07-01

**Clarity:** 3
**Significance:** 2
**Originality:** 2
**Rating:** 5
**Confidence:** 3

**Summary:**

The authors introduce the Opinion Interval Representation problem: given a signed graph, assign each node to an interval on the real line in a way that maximizes the number of positive edges between nodes whose intervals overlap and the number of negative edges between nodes whose intervals are disjoint. They show the problem is NP hard and relate it to Interval Editing and Correlation Clustering. The authors provide a PTAS for the case when only $k$ distinct intervals are allowed and two greedy and simulated annealing heuristics. In experiments on real-world signed graph datasets, the authors find that interval representations found by their heuristics are more expressive than Correlation Clustering solutions. In synthetic data, they find that their heuristics can recover true interval structure. Finally, in a dataset of German parliamentary votes, the authors show one-dimensional ideologies of politicians can be recovered using overlapping intervals.

**Questions:**

I don't have any major questions, but I do have the following suggestions/comments:

1. I don't understand Figure 3... It introduces many symbols not mentioned in the text (S, M, T, $H_S$, ...) and it's not at all clear how this relates to a signed graph. The text/caption needs to provide enough detail for the reader to gain something from the figure (if there isn't enough space, it should be in the appendix).
2. I think it's important to make clear that GAIA and VENUS are not finding better Correlation Clustering solutions in Table 1: the 8-chain structure is intrinsically more forgiving than clustering, since edges between nodes assigned to an interval and those on either side are not penalized. (i.e., the authors are assigning 2-3 cluster labels to each node and counting agreement if any labels match between nodes connected by a positive edge). In particular, "our novel methods outperform state-of-the-art heuristics for CORRELATION CLUSTERING. On 8 real-world datasets, the results are at least 20% better and over 38% on average" is misleading--they're being compared on two different tasks.

Minor:
"in recent years we are increasingly witnessing conflicts based on different views and opinions." is this really a recent phenomenon? Seems like people have been disagreeing since Plato and Aristotle (and earlier...). I don't think the motivation needs to rely on an unbacked claim that conflict is increasing (I would add sources or cut the claim).

**Ethical Concerns:**

["NO or VERY MINOR ethics concerns only"]

**Final Justification:**

The updates described by the authors address my concerns about presentation. I've updated my score accordingly.

**Limitations:**

The authors are clear about the extent of their results, and I don't see any potential for negative societal impact.

**Quality:**

3

**Strengths And Weaknesses:**

Strengths:
1. The new algorithmic problem is explored thoroughly, with hardness, approximation, and heuristics.
2. The experiments do a good job of showcasing the problem and the heuristics

Weaknesses:
1. The experimental comparison to Correlation Clustering is presented in a misleading way in certain places

Quality: Seems high throughout the paper.  Good grounding in existing work and a well-executed paper overall.

Clarity:
The paper is generally well-written, with only a few unclear points (see suggestions below.)

Significance: I think this paper would be of interest to the the community interested in signed networks, community detection, and ideology estimation.

Originality: This paper explores a novel problem (to my knowledge), but the techniques are standard. Another piece of related work in the line of recovering ideological positions of legislators from co-voting data is the (DW-)NOMINATE algorithm of Poole and Rosenthal. That method has the advantage of providing continuous estimates for each politician rather than an interval representation.

---

> ### Author Rebuttal · Authors · 2025-07-30
>
> We thank the reviewer for the positive evaluation of our paper.
>
> > I think it's important to make clear that GAIA and VENUS are not finding better Correlation Clustering solutions in Table 1: the 8-chain structure is intrinsically more forgiving than clustering, since edges between nodes assigned to an interval and those on either side are not penalized. (i.e., the authors are assigning 2-3 cluster labels to each node and counting agreement if any labels match between nodes connected by a positive edge). In particular, "*our novel methods outperform state-of-the-art heuristics for CORRELATION CLUSTERING. On 8 real-world datasets, the results are at least 20% better and over 38% on average*" is misleading--they're being compared on two different tasks.
>
> We will rephrase this to ensure that there are no misunderstandings. In particular, we will reword the cited part to: "*In our experiments, we compare our novel methods against state-of-the-art heuristics for CORRELATION CLUSTERING, which is a special case of our problem. On 8 real-world datasets, our methods using overlapping opinion intervals achieve an error at least 20% better and over 38% on average, compared to the methods solving (the less expressive) CORRELATION CLUSTERING problem.*"
>
> We will also make sure that there are no such misunderstandings in the rest of the paper.
>
> As a side note, we remark that our algorithms also perform surprisingly well when applied directly to correlation clustering. In the supplementary material in Table 4, we evaluated GAIA and VENUS on solving correlation clustering instances (i.e., they are given 8 non-overlapping intervals as input which corresponds to solving correlation clustering with 8 clusters). Several times they match the performance of the best correlation clustering heuristics we compare against, and their error is never more than 2% higher than the best dedicated method for correlation clustering.
>
> > I don't understand Figure 3... It introduces many symbols not mentioned in the text ($S$, $M$, $T$, $H_S$, ...) and it's not at all clear how this relates to a signed graph. The text/caption needs to provide enough detail for the reader to gain something from the figure (if there isn't enough space, it should be in the appendix).
>
> Indeed, in the main text we only mention the vertices $L$ and $R$ from the reduction, but not the auxiliary vertices $S$, $M$, $T$, $H_S$ and $H_T$, which are necessary for our reduction (see Section A.1). We will move the figure to the appendix for the final version of the paper.
>
> > Minor: "in recent years we are increasingly witnessing conflicts based on different views and opinions." is this really a recent phenomenon? Seems like people have been disagreeing since Plato and Aristotle (and earlier...). I don't think the motivation needs to rely on an unbacked claim that conflict is increasing (I would add sources or cut the claim).
>
> We will rephrase this to state that: "*in recent years there has been a lot of research to understand the conflicts in social networks and how they are based on different views and opinions.*" We will do this in the abstract, as well as at the start of the introduction, where we will also add some citations for this claim.
>
> > Another piece of related work in the line of recovering ideological positions of legislators from co-voting data is the (DW-)NOMINATE algorithm of Poole and Rosenthal. That method has the advantage of providing continuous estimates for each politician rather than an interval representation.
>
> Thank you for pointing out this work. We will cite it and discuss it in the final version of our paper. However, we stress that this work is only applicable for co-voting behavior in the case study that we constructed for the German Bundestag, where the available data consisted of politicians and how they voted on parliamentary bills. In other words, the DW-NOMINATE method requires a signed bipartite graph, with the two sides given by politicians and parliamentary bills. In contrast, our algorithm takes as input a unipartite graph only consisting of the politicians and whether they frequently co-voted or not. Thus, in such settings the DW-NOMINATE algorithm is not applicable to the best of our knowledge.

---

> > ### Comment · Reviewer_NU77 · 2025-08-01
> >
> > Thanks for the replies! Those updates sound good overall. However, I still worry that the updated sentence makes it sound like your algorithms are performing better than the correlation clustering algorithms:
> >
> > > In our experiments, we compare our novel methods against state-of-the-art heuristics for CORRELATION CLUSTERING, which is a special case of our problem. On 8 real-world datasets, our methods using overlapping opinion intervals achieve an error at least 20% better and over 38% on average, compared to the methods solving (the less expressive) CORRELATION CLUSTERING problem.
> >
> > The real comparison seems to be between the expressivity of opinion interval representation and correlation clustering, as is made clear in Section 4 (but not in the introduction sentence). I would write something like the following to make this crystal clear:
> >
> > > Our experiments find that OPINION INTERVAL REPRESENTATION is substantially more expressive than CORRELATION CLUSTERING and that our algorithms succeed in exploiting this expressivity. On 8 real-world datasets, our methods for find overlapping opinion intervals that represent the data with 38% fewer disagreements on average compared to CORRELATION CLUSTERING solutions found by state-of-the-art methods.
> >
> > With the updates described in the rebuttal, I'd be happy to see this paper accepted and will update my score accordingly.

---

> > > ### Author Response · Authors · 2025-08-04
> > >
> > > Thank you for your prompt response and for increasing your score. We believe the sentence you have suggested is very clear and we will use in the introduction of the final version of our paper.

---

### Official Review · Reviewer_SMfw · 2025-07-02

**Clarity:** 4
**Significance:** 3
**Originality:** 3
**Rating:** 5
**Confidence:** 2

**Summary:**

The manuscript's main contribution is the opinion interval representation, a framework for analyzing signed graphs. The proposed method assigns intervals on the real line to nodes, maximizing overlap for positively connected pairs and disjointness for negatively connected pairs (see Section 2). This method differs from correlation clustering, which partitions nodes into disjoint clusters. Another contribution of the paper is establishing NP-hardness for general instances (see Section 2.1). It also provides a polynomial-time approximation scheme (PTAS) for a restricted complete-graph scenario with a fixed number of intervals (see Section 2.2). Finally, the manuscript presents two scalable heuristics, GAIA and VENUS (see Section 3), for inference in the interval representation. Experiment results are encouraging and outperform state-of-the-art correlation clustering approaches on both synthetic and real-world datasets.

**Questions:**

* How should practitioners select the amount and overlap configuration for the intervals in GAIA and VENUS? Let us assume no domain expertise with the new datasets (so no reasonable guess can be made for these values). Are there any principled or data-driven approaches for this task?
* Could the authors discuss potential extensions of this method to multi-dimensional interval representations? Although outside the scope of this work, a discussion on the challenges of higher-dimensional interactions, including their real-world applicability, would enhance the significance of this framework.

**Ethical Concerns:**

["NO or VERY MINOR ethics concerns only"]

**Final Justification:**

After reviewing the responses from my peer reviewers and the authors, I believe this paper should be accepted. The other reviews are positive and highlight relevant strengths of the work. The authors have appropriately addressed my question, including my suggestion for multi-dimensional extension.

**Limitations:**

Yes

**Quality:**

3

**Strengths And Weaknesses:**

**Strengths**:

* Novelty and a principled approach to a solution. The proposed framework for opinion interval representation allows for higher expressivity when modelling real-world scenarios (as shown in Figure 1 in the German parliament case study). Moreover, the theoretical results in Section 2 establish clear boundaries for the application of this framework. For instance, Theorem 2.3 provides grounding for an approximate solution, given that Section 2.1 shows that the problem is NP-hard.
* Experiment results are thorough and convincing. The manuscript compares the method with four SOTA baselines, including a novel dataset based on voting data from the German Parliament. Table 1, which compares the proposed heuristics with the baseline, achieves an improvement of 20% to 90%.

**Weaknesses**:

* The polynomial-time approximation in Theorem 2.3 could limit the method's application due to graph completeness and fixed interval numbers. It is unclear from the manuscript the impact and size of practical limitations on real-world cases.
* The hyperparameter selection for GAIA and VENUS could be challenging for some applications. The manuscript does not offer a data-driven approach for selecting the number of intervals or the overlap configuration.

---

> ### Author Rebuttal · Authors · 2025-07-30
>
> We thank the reviewer for the positive evaluation of our paper.
>
> > How should practitioners select the amount and overlap configuration for the intervals in GAIA and VENUS? Let us assume no domain expertise with the new datasets (so no reasonable guess can be made for these values). Are there any principled or data-driven approaches for this task?
>
> In such cases, we recommend running the algorithms with multiple interval structures, varying both the number of intervals and their overlap. Then, one can pick the one with the lowest objective function value. Here, two important remarks are in order:
> * More intervals will generally be more expressive (leading to better objective values) but less interpretable. To find a good tradeoff, one can adopt methods such as the elbow method from the clustering literature.
> * Regarding how to pick the intervals and their overlap structure, we can exploit the fact that our algorithms are highly efficient.
> In particular, due to this efficiency we can enumerate or randomly sample a large number of interval structures and then one can pick the one with the best objective function value. Further, the results that we reported for Reviewer HyVH with varying number of intervals suggest that typically a small number of intervals is sufficient (in this particular case, we observed that for $k$-chains, the objective function values only improve slightly for $k>8$; this is also in line with results from the correlation clustering literature), limiting the search space further.
>
> Additionally, given that our algorithms are iteration-based and converge quickly, one could speed up the procedure above using methods from the hyperparameter search literature. Specifically, low-budget Bayesian optimization methods like *successive halving* by Jamieson and Talwalkar (citation below) should be well-suited to select promising configuration candidates quickly. Here, a configuration consists of an interval structure and the hyperparameters of the algorithms. Now, we run our algorithms for a small number of iterations for many different configurations. Then, the successive halving procedure will be able to pick the most promising candidates automatically to use more iterations on them. This is done multiple times until, at the end, we find one most promising solution.
>
> We will add a discussion of this in the appendix of the final version of our paper.
>
> Kevin G. Jamieson, Ameet Talwalkar: Non-stochastic Best Arm Identification and Hyperparameter Optimization. AISTATS 2016: 240-248
>
> > Could the authors discuss potential extensions of this method to multi-dimensional interval representations? Although outside the scope of this work, a discussion on the challenges of higher-dimensional interactions, including their real-world applicability, would enhance the significance of this framework.
>
> Thank you for raising this interesting point. Surprisingly, after discussing the reviewer's suggestion, we realized that all of our algorithms (both the heuristics, as well as the FPT algorithm) also extend to higher-dimensional settings without any significant changes:
> * The assignment-procedure from the heuristic GAIA (Algorithm 1) can be readily adapted, because it only uses the overlap-structure of the geometric objects.
> * Similarly, this is also the case for the analysis of the FPT algorithm, which works with overlapping clusters and this analysis thus also extends when we have higher-dimensional objects.
> We will mention this in the final version of the paper, and we will also mention that studying this empirically appears to be an interesting question for future work.

---

### Official Review · Reviewer_HyVH · 2025-07-03

**Clarity:** 3
**Significance:** 3
**Originality:** 4
**Rating:** 5
**Confidence:** 2

**Summary:**

This paper introduces a generalization of correlation clustering, which they call the “Opinion Interval Representation” problem. The idea is that we are given a set of nodes and a set of signed edges. In correlation clustering, the goal is to assign nodes to clusters in such a way that we minimize the total number of negative edges within a cluster plus the number of positive edges between two different clusters. This paper consider instead assigning each node to an interval on the real line. If two nodes have overlapping intervals, we recieve a penality if they have a negative edge between them. If nodes have non-overlapping intervals, they receive a penalty if connected by a positive edge. The idea is to choose intervals to minimize the number of penalties.

The justifcation for this problem is that we might have non-transitive edge forming behavior in social networks — the authors give the compelling example of politicians — a centrist candidate might often vote the same way as both a liberal candidate and a conservative candidate, but those candidates rarely vote in the same way. The authors also provide some nice experimental justification — they apply the clustering approach to a new data set based on German Parliment votes, and recover a natural picture of the political “spread” of different parties. They also show that, on all examples datasets they test, the problem leads to significanty better “explanations” than correlation clustering -- i.e., it is possible to produce an interval representations that gets many more edges correct than a regular clustering. This is despite the fact that these experiments actually solve a restricted version of the problem, where we have a fixed number of intervals k (they choose k = 8 for some reason) and assign every node to one of those intervals.

The theoretical work of the paper is mainly focused on algorithms, and read more like an algorithms conference paper than a machine learning paper. The authors first prove the problem as stated is NP-hard, which is not too surprising. They then describe a fixed-parameter tractible algorithm for their restricted version of the problem involving a fixed set of k intervals — the runtime is polynomial but exponential in k. This method is based on adapting an FPT algorithm for correlation clustering when the number of clusters is fixed (in general, correlation clustering allows any number of clusters — the idea is to find the least disagreements amoung all possible partitions). Basically, but brute forcing over all possible “overlaps” of the fixed set of intervals, they can reduce to a generalization of correlation clustering where disagreement between some clusters is not penalized.

The authors also describe some heuristic algorithms for the problem, which they test in an experiments section. The experiments section is complete in that it asks clear questions and tells a clear story: the authors justify they their new problem is interesting (it’s more expressive than correlation clusteirng for real dataset), that their heuristics find reasonable solutions in a reasonable amount of time, and that sovling the problem can product interesting visualizations/explanations (via their German parliment example).

Small comments:
- It doesn’t seem necessary to write k = O(1) on Page 2 since you are being explicit about the dependence on k in the stated runtime.
- In the introduction it is mentioned that the paper approach leads to results that are “20% better than correlation clustering”. It would be helpful to describe here roughy what metric is being considered — how methods for two different problems compared in a head-tohead way?
- A few times in the intro is was mentioned that something would be discussed later or “below” but I would have preferred a reference to exactly where. E.g.: Below, we will also argue that Interval Editing problems are closely related. Some of these points I also didn’t think were worth delaying — it is pretty obvious why the problem is more expressive than correlation clustering — could be explained in 2 sentences instead of defering until Section 2.
- The name "OPINION INTERVAL REPRESENTATION” makes me think of the problem of finding an *exact* representation if one exists, but Problem 2.1 is more general — maybe a better name would be something like “best interval approximation” or something?
- The text claims that OPINION INTERVAL REPRESENTATION “is not FPT when parameterized by the number of required edge deletions”, but the problem doesn’t even involve edge deletions at all. I think it would be better (if it is correct) to say when parameterized by OPT where OPT is the minimum cost solution of OPINION INTERVAL REPRESENTATION? Or at least to explain things more.
- Why in the experiments did you restrict to exactly 8 intervals?
- Should there be a last line of Theorem 2.2 that drives the point home, e.g. “Thus, the Opinion Interval Representation problem is NP-hard”.

**Questions:**

I listed maybe one small question with my summary, and a few other small suggestions, but honestly I don't have any major suggestions. I thought the paper was well written and an interesting read.

**Ethical Concerns:**

["NO or VERY MINOR ethics concerns only"]

**Final Justification:**

No major negative points about the paper were raised by other reviewers, so my feelings on the paper remain positive for the reasons described in my review.

**Quality:**

4

**Strengths And Weaknesses:**

While I am not an expert on the topic, this feels like a very complete paper to me. The paper is well-written and easy to follow. The authors introduce a natural and easy to understand clustering objetive. Given its connection to correlation clustering, and how well interval graphs have been studied, the problem is not too far afield from prior work. At the same time, it appears significantly more expressive than correlation clustering, while still resulting in an easy to interpret visualization for structure that could possibly have lead to the observed signed graph (k 1-d intervals instead of k clusters). At least this is true of the restricted version of the problem. While none of the theoretical results in the paper are ground breaking, the authors take meaningful first steps towards understanding algorithms for their objective, and applying it to a case study. Perhaps the only “negative” I can see for this paper is that it’s not an amazing fit for a machine learning conference, although correlation clustering seems to have been studied quite a bit in the ML community.

---

> ### Author Rebuttal · Authors · 2025-07-30
>
> We thank the reviewer for the positive evaluation of our paper. We will address all of the small comments in the final version and comment on some of them below.
>
> > Why in the experiments did you restrict to exactly 8 intervals?
>
> We decided to use $k=8$ intervals because in our experiments this provided the best tradeoff between interpretability and good objective function values. To illustrate this, we provide a table below showing the number of violated edges by GAIA and VENUS for different values of $k=4,8,12,16$. We report the best result achieved by the algorithm across 50 runs.
>
> | instance     | algorithm |  best_4 | best_8 | best_12 | best_16 |
> | :----------- | :-------- | ------: | -----: | ------: | ------: |
> | Bitcoin      | GAIA      |     768 |    711 |     721 |     725 |
> | Bundestag    | GAIA      |   3,772 |  1,001 |   1,030 |   2,584 |
> | Chess        | GAIA      |   7,279 |  6,472 |   6,469 |   6,467 |
> | Epinions     | GAIA      |  34,213 | 31,687 |  31,959 |  31,805 |
> | Slashdot     | GAIA      |  51,367 | 45,120 |  44,860 |  44,827 |
> | WikiConflict | GAIA      | 140,004 | 69,344 |  68,094 |  67,761 |
> | WikiElec     | GAIA      |  11,937 | 11,275 |  11,265 |  11,300 |
> | WikiSigned   | GAIA      |  38,513 | 35,204 |  35,003 |  35,008 |
> | Bitcoin      | VENUS     |     819 |    760 |     750 |     716 |
> | Bundestag    | VENUS     |   3,819 |  1,001 |   1,002 |  11,704 |
> | Chess        | VENUS     |   7,276 |  6,410 |   6,426 |   6,398 |
> | Epinions     | VENUS     |  34,112 | 31,286 |  31,287 |  31,445 |
> | Slashdot     | VENUS     |  51,117 | 44,563 |  44,447 |  44,462 |
> | WikiConflict | VENUS     | 142,450 | 69,014 |  67,645 |  67,472 |
> | WikiElec     | VENUS     |  11,923 | 11,297 |  11,302 |  11,255 |
> | WikiSigned   | VENUS     |  38,083 | 34,564 |  34,819 |  35,062 |
>
> The results show that up to $k=8$ intervals, the objective function values still decrease substantially, and afterwards they only decrease slightly. We note that for the Bundestag dataset, the results for $k=16$ are worse because for this dataset the variance of the algorithms' solutions is larger; with more runs we could also obtain better results here and we stress that this is not the case for any other datasets.
>
> We will add this table, as well as plots for selected datasets, in the final version of the paper.
>
> If the reviewer wishes, we can also provide the results for averages of the objective function values (rather than report the best results) for all datasets.
>
> > Perhaps the only “negative” I can see for this paper is that it’s not an amazing fit for a machine learning conference, although correlation clustering seems to have been studied quite a bit in the ML community.
>
> We submitted the paper to NeurIPS because we believe its community brings together researchers with interest in correlation clustering, opinion formation, as well as signed graphs. Therefore, we think the paper is a good fit. To make this more explicit, we will cite the papers listed below in the final version of our paper, which highlight the community's interest in these topics.
>
> Yuko Kuroki, Atsushi Miyauchi, Francesco Bonchi, Wei Chen: Query-Efficient Correlation Clustering with Noisy Oracle. NeurIPS 2024
>
> Sepehr Assadi, Vihan Shah, Chen Wang: Streaming Algorithms and Lower Bounds for Estimating Correlation Clustering Cost. NeurIPS 2023
>
> Konstantin Makarychev, Sayak Chakrabarty: Single-Pass Pivot Algorithm for Correlation Clustering. Keep it simple! NeurIPS 2023
>
> Silvio Lattanzi, Benjamin Moseley, Sergei Vassilvitskii, Yuyan Wang, Rudy Zhou: Robust Online Correlation Clustering. NeurIPS 2021: 4688-4698
>
> Ruo-Chun Tzeng, Bruno Ordozgoiti, Aristides Gionis: Discovering conflicting groups in signed networks. NeurIPS 2020
>
> Yanbang Wang, Jon M. Kleinberg: On the Relationship Between Relevance and Conflict in Online Social Link Recommendations. NeurIPS 2023
>
> Liwang Zhu, Qi Bao, Zhongzhi Zhang: Minimizing Polarization and Disagreement in Social Networks via Link Recommendation. NeurIPS 2021: 2072-2084
>
> > The name "OPINION INTERVAL REPRESENTATION” makes me think of the problem of finding an exact representation if one exists, but Problem 2.1 is more general — maybe a better name would be something like “best interval approximation” or something?
>
> Thank you for raising this point, indeed the current name could suggest that we are looking for an exact representation. As suggested by the reviewer, we will rename the problem to *best interval approximation* in the final version of our paper.
>
> > In the introduction it is mentioned that the paper approach leads to results that are “20% better than correlation clustering”. It would be helpful to describe here roughly what metric is being considered — how methods for two different problems compared in a head-tohead way?
>
> We will clarify this in the final version of our paper, stating that we evaluate the algorithms on our objective function and that correlation clustering algorithms are more restricted when searching for solutions, since correlation clustering is less expressive than our problem. We will also implement all of the related changes that we discuss below for Reviewer NU77.

---

> > ### Comment · Reviewer_HyVH · 2025-08-04
> >
> > Thanks for the response. I am planning to keep my positive evaluation of the work.

---

### Note · Authors · 2025-08-13

We thank the reviewers for their constructive feedback and the positive evaluation of our paper. We have nothing to add to the discussion, and use this opportunity to summarize the changes that we will incorporate into the final version of our paper:

- We will include in the appendix a table comparing the number of violated edges in solutions found by GAIA and VENUS for k-chains with different values of k=4,8,12,16.

- We will expand the introduction to cite additional relevant papers that highlight the community's interest in the topics addressed by our work. We will also cite (DW-)NOMINATE and discuss its applicability in the Bundestag case study, where input can be given as a bipartite graph.

- Following Reviewer HyVH’s suggestion, we will rename the problem to *Best Interval Approximation* to clarify that we are not seeking an exact representation, but rather a representation that maximizes agreement.

- We will make sure there are no misunderstandings regarding the performance of GAIA and VENUS and state-of-the-art correlation clustering algorithms. In particular, we will adopt Reviewer NU77’s suggestion and clarify that the difference in the error is due to the increased expressivity of Best Interval Approximation, which our algorithms manage to exploit.

- We will add a new appendix section detailing the selection of interval structures and hyperparameters for GAIA and VENUS.

- We will remark on the extension to multi-dimensional intervals and mention an empirical study as an interesting direction for future work.

- We will move Figure 3 to the appendix, where the auxiliary vertices are explained.

- We will further address all other small comments as mentioned in the discussion.

---

### Decision · Program_Chairs · 2025-09-17

**Decision:**

Accept (oral)

**Comment:**

This work examines the problem of inferring opinion ranges of nodes in a signed network (where edges have positive or negative signs corresponding to agreement or conflict between the corresponding nodes). In this optimization problem, we seek to assign opinion intervals (i.e., acceptable range of opinions) to each node such that two nodes connected by a positive edge should have overlapping intervals and those connected by a negative edge should have disjoint intervals. This is in contrast to assigning nodes to clusters so as to maximize the number of edges consistent with the clustering.

The reviewers praised several strengths of this work, including:
* A compelling and clearly described problem, as well as easy-to-follow writing, ample examples for building intuition, and clear presentation of results
* An intuitive formulation of this problem (i.e., by considering opinion ranges), which creates a clean mathematical formulation and is also grounded in real world observations.
* Reasonable theoretical results which add to the paper's comprehensiveness. Though many of these theoretical results are not very surprising and follow a similar flow to other related works, they can serve as a foundation for future work. There could be interesting follow up papers considering a dynamic setting where opinion ranges expand or contract over time. Similarly, dropping the independence assumptions where, e.g., nodes have "innate" opinion ranges that expand or contract depending on the nodes' centrality or the homophily of its immediate neighborhood.
* An interesting dataset, which, to my knowledge, may be new to the networks community, and gives a compelling set of empirical insights.

The reviewers provide detailed questions and local suggestions for improvement. We thank the authors for their engagement through the discussion process and for sharing the list of edits they plan to make in their final remark. Though space is tight, we encourage the authors to also add a short discussion and suggestions for future work, including some learning-based questions to encourage engagement and follow-up work.